

# Storage and routing of water in the deep critical zone of a snow dominated volcanic catchment

Alissa White[1], Bryan Moravec[2], Jennifer McIntosh[1], Yaniv Olshansky[2], Ben Paras[1], R. Andres Sanchez[1], Ty P. A. Ferré [1], Thomas Meixner[1], Jon Chorover[2]

[1]Department of Hydrology and Atmospheric Sciences, University of Arizona, Tucson, 85721, USA

[2]Department of Soil, Water, and Environmental Sciences, University of Arizona, Tucson, 85721, USA

*Correspondence to*: Alissa White (alissawhite@email.arizona.edu)




**Abstract.** This study combines major ion and isotope chemistry, age tracers, fracture density characterizations, and physical hydrology measurements to understand how the structure of the critical zone (CZ) influences its function, including water routing, storage, mean water residence times, and hydrologic response. In a high elevation rhyolitic tuff catchment in the Jemez River Basin Critical Zone Observatory (JRB-CZO) within the Valles Caldera National Preserve of northern New Mexico, a periodic precipitation pattern creates different hydrologic flow regimes during
spring snowmelt, summer monsoon rain, and fall storms. Hydrometric, geochemical, and isotopic analyses of surface water and groundwater from distinct stores, most notably a perched aquifer in consolidated collapse breccia and deeper groundwater in a fractured tuff aquifer, enabled us to untangle the interactions of these groundwater stores and their contribution to streamflow across one complete water year.

Despite seasonal differences in groundwater response due to water partitioning, major ion chemistry indicates
that deep groundwater from the highly fractured site is more representative of groundwater contributing to streamflow across the entire water year. Additionally, comparison of streamflow and groundwater hydrographs indicates hydraulic connection between the fractured welded tuff aquifer and streamflow while the perched aquifer within the collapse breccia deposit does not show this same connection. Furthermore, analysis of age tracers and stable water isotopes indicates that groundwater is a mix of modern and older waters recharged from snowmelt and downhole neutron probe
surveys suggest that water moves through the vadose zone both as vertical infiltration and subsurface lateral flow, depending on lithology. We find that in complex geologic terrain like that of the JRB-CZO, differences in CZ architecture of two hillslopes within a headwater catchment control water storage and routing through the subsurface and suggest that the perched aquifer does not contribute significantly to streams while deep fractured aquifers contribute most to streamflow.

## 1   Introduction

Understanding the interconnections of groundwater and surface water is fundamental to water resource management as groundwater and surface water should be considered a single resource (Winter, 1998); however, their
interactions in different hydrogeologic settings are varied and complex (Winter, 1999). Discerning stream water sources and groundwater dynamics are even more important in the context of changing climate, especially in the semiarid, mountainous environment of the western United States where warming trends are expected to threaten water supply (Barnett et al., 2005). Furthermore, identifying compartmentalized groundwater stores is necessary to sufficiently account for all components of the water balance (McDonnell, 2017). Therefore, characterizing localized
water stores and the hydrologic connection of those aquifers to streams in mountainous environments that act as water towers (Viviroli et al., 2007) has important implications for water resource availability of large population centers downstream.

The influence of the hydrogeologic environment (i.e. geology, topography, and climate) on the groundwater flow system of a given geographic region has long been accepted as the theoretical framework used to conceptualize
groundwater flow. Building on Toth's (1970) conceptual model and the understanding that one part of the framework informs our knowledge of the other, several studies have focused on topography (Beven and Kirkby, 1979; Woods et al., 1997; Hutchinson and Moore, 2000; Kirchner et al., 2001; McGuire et al., 2005) and rock type (Farvolden, 1963;





Freeze, 1972; Kelson and Wells, 1989; Mwakalila et al., 2002) as controls of groundwater flow systems. However, subsurface heterogeneities, which can be abundant and are challenging to identify, can give rise to complex, localized

groundwater stores, whose contribution to streamflow can be very difficult to discern. There is still much to learn about the extent to which structural heterogeneities exist and how, specifically, they control groundwater storage, routing, and contributions to streamflow. For instance, evidence of perched aquifers transmitting shallow subsurface flow has been shown across variable rock types (Salve et al., 2012; Kim et al., 2014; Brantley et al., 2017; Kim et al., 2017; McIntosh et al., 2017). Furthermore, Brooks et al. (2015) highlighted the need to understand the influence of

subsurface structure on water routing and residence time as they concluded that surface water, across several catchments and flow regimes, substantially interacted with or spent time within various soil and groundwater reservoirs.

Heavily instrumented and intensively studied sites, such as Critical Zone Observatories (CZOs), which are part of a network of field-based laboratories arrayed across a variety of rock types, land uses, elevations and climates (Anderson et al., 2008), are ideal locations to examine the interplay between subsurface structure and function.

Moreover, recent focus on characterizing the deep subsurface architecture is beginning to elicit a deeper understanding of the role of weathering, lithology, and hydrology in overall CZ function and landscape evolution (Riebe et al. 2017). The critical zone (CZ) is the near-surface terrestrial layer of the Earth that spans from the tops of trees down to unweathered bedrock where water, rock, air, and life meet and interact (Brantley et al., 2006, 2007; Anderson et al.,

2007; Chorover et al., 2007; Kusel et al., 2016). Understanding the coupling between CZ architecture, developed over geologic time scales, and CZ function on short event time scales is a primary goal of CZ science (Chorover et al., 2011; Brooks et al., 2015). In particular, there is limited knowledge about the structure of the deep CZ and its direct influence on water storage (Holbrook et al., 2014; Dralle et al., 2018) and routing, mean residence times (McGuire et al., 2005), and streamflow sources. Furthermore, Wlostowski (In Review) notes in a cross-CZO study that the lack of

subsurface characterization hinders our ability to relate catchment structure and hydrologic behavior in meaningful ways. Integrated studies that simultaneously examine both, CZ architecture and CZ hydrology, through hydrometric, geophysical, geochemical, and residence time analyses are needed to understand the distribution of groundwater stores, their connection to streamflow, and the underlying impact of CZ architecture on hydrologic response to climatic drivers.

A current focus of hydrology is quantifying and predicting groundwater storage (Holbrook et al., 2014; McDonnell, 2017, Rempe and Dietrich, 2018; Dralle et al., 2018; Bhanja et al., 2018; Wlostowski et al., In Review) and geophysics is an important tool for examining CZ architecture and its influence on water storage and movement. For example, McGuffy (2017) used seismic refraction surveys to estimate porosity and found that initial porosity plays a significant role in bedrock weathering in granitic and rhyolitic tuff CZs. Flinchum et al. (2018) took those porosity

calculations a step further in using geophysics to estimate the water holding capacity of another granitic CZ; however, both studies noted the strong influence of, and uncertainty associated with degree of saturation of the media. Rempe and Dietrich (2018) used downhole surveys with a neutron probe to estimate rock moisture in the CZ and Dralle et al. (2018) used geophysics-based storage estimates from Rempe (2016) and Rempe and Dietrich (2018) to suggest differences in direct and indirect storage within the CZ from a coupled mass balance and storage-discharge function.



The complexity of these estimates and their interactions highlights the need to couple geophysical approaches with subsurface interrogation, such as drilling and field characterization of hydraulic properties, to resolve this complexity, particularly in fractured heterogeneous environments.

In a headwater catchment and nested zero order basin (ZOB) within the complex volcanic Jemez River Basin Critical Zone Observatory (JRB-CZO), a considerable amount of research has been done to characterize the hydrology

of the system. For instance, previous studies have explored energy limitations and topographic controls on hydrologic partitioning and water transit times (Zapata-Rios et al., 2015a, 2015b). Other studies used carbon pool and rare earth elements and ytrrium (REY) as biogeochemical tracers of streamflow generation (Perdrial et al., 2014; Vazquez-Ortega et al., 2015, 2016) and estimated groundwater contributions using end member mixing analyses (Liu et al., 2008a, 2008b). The most recent JRB-CZO studies explored concentration-discharge relationships to study seasonal

shifts of hydrologic flow paths (McIntosh et al., 2017) and identify the hydrochemical processes governing the transport behavior of five distinct groups of solutes (Olshansky et al., 2018). Furthermore, studies agree that there is little overland flow contribution to streamflow in headwater catchments (Liu et al., 2008b; Perdrial et al., 2014; Zapata-Rios et al., 2015a) and subsurface flow is the primary contributor to streamflow (Liu et al., 2008a, 2008b; Perdrial et al., 2014; Vazquez-Ortega et al., 2015; Zapata-Rios et al., 2015a; McIntosh et al., 2017; Olshansky et al., 2018;

Wlostowski et al., In Review).

Studies spanning several water years have shown that spring snowmelt and summer monsoons induce different surface water flow regimes. More specifically, groundwater recharge appears to be restricted to winter snowmelt (McIntosh et al., 2017) and large evaporative fluxes diminish streamflow in summer months (Zapata-Rios et al., 2015a). However, seasonal groundwater changes have not been previously observed here and the interaction of

different stores of water within the subsurface and the timing of their connection to streamflow are not understood. This gap motivated the current study, which sought to relate groundwater response, geochemistry, and age tracers across a full water year to the characterization of subsurface structure, mineralogy, and hydraulic properties. We hypothesized that there will be a more dramatic hydrologic response of shallow groundwater to spring snowmelt and a more gradual, small change after summer monsoon events. This study also aimed to elucidate how multiple

groundwater stores within the CZ contribute to streamflow during different seasonal hydrologic flow regimes.

Most upland catchment studies to date have used springs as proxies for groundwater in the JRB-CZO; however, recent work by Frisbee et al. (2013) showed that while groundwater is a significant component of most springs, no springs are consistently composed entirely of groundwater. With the recent drilling of a set of nested monitoring wells in a headwater catchment at the JRB-CZO (Figure 1B), we can now directly access groundwater from several depths

within the CZ (Figure 2). This enabled the geochemical and isotopic analysis of groundwater and surface water from the JRB-CZO to answer the following research questions:

1) What is the seasonal hydrologic response of groundwater as a function of depth below ground surface in two hillslopes with contrasting lithology and CZ architecture?

2) How does CZ architecture, such as fracture density, lithology, and mineralogy control seasonal groundwater

contribution to streams?



To address these questions, we integrated several types of analyses including hydrometric, geophysical, geochemical, isotopic, and residence time tracers to examine the hydrologic response of ground and surface water and understand the connection between distinct groundwater stores and streamflow. We compared the timing of streamflow and groundwater response to climatic drivers, quantified temporal changes in subsurface water storage, defined distinct groundwater stores, inferred recharge processes from stable water isotopes and age tracers, and examine how local flow processes relate to larger scale patterns.

## 2    Study Site and Methods
### 2.1 Site description

The Jemez River Basin Critical Zone Observatory (JRB-CZO) within the Valles Caldera National Preserve is situated in the Jemez Mountains in northern New Mexico northwest of Albuquerque (Figure 1A). This region is located in the transition zone between the snow dominated Rocky Mountains and the North American Monsoon (NAM) dominated deserts of the southwestern United States (Broxton et al., 2009). The JRB-CZO is in a montane, continental, sub-humid to semiarid climate characterized by a bimodal precipitation pattern (Zapata-Rios et al., 2015b). The VCNP is in a 21 km wide caldera that formed approximately 1.25 Myr (Bailey et al., 1969; Self et al., 1986). Ongoing volcanic activity as recent as 40 kyr caused the uplift of several resurgent domes throughout the caldera (Wolff et al., 2011) of which Redondo Peak (3,432 meters above sea level, masl) is the largest. The JRB-CZO comprises several headwater catchments that drain different aspects of Redondo Peak. The geology of Redondo Peak is characterized by several faults and is dominated by Pleistocene aged Bandelier Tuff, rhyolite, and andesitic rocks (Broxton et al., 2009; Vazquez-Ortega et al., 2015) that were intermixed by collapse breccias in some locations (Hulen and Nielson, 1991), which created highly heterogeneous and complex geology (Figure 1B).

La Jara catchment and the Mixed Conifer Zero Order Basin (referred to as ZOB from here on), which is nested within the headwaters of La Jara, drain the eastern side of Redondo Peak (Figure 1B). La Jara catchment ranges in elevation from 2,702 to 3,429 masl with a mean slope of 15.7° and drains an area of 2.66 km$^2$ (Perdrial et al., 2014). The ZOB consists of SE- and SW- facing slopes that are separated by a convergent zone and drains 0.15 km$^2$ (Vazquez-Ortega et al., 2016). The convergent zone of the ZOB, just above the ZOB flume, is characterized by boggy land in which standing water is present during the wettest parts of the year. Further evidence of near surface saturation in this area is the presence of marshy plants in standing water areas (e.g. broad-leaf cattails [*Typha latifolia*], rocky mountain irises [*Iris missouriensis*] and skunk cabbage [*Veratrum californicum*]), which contrast with those in the surrounding upland area of the ZOB (mostly bunch grasses [e.g. *Deschampsia, Festuca, Koeleria, Muhlenbergia,* and *Poa*] and aspen [*Populus tremuloides*] regrowth following a wildfire in 2013). Surface water from the ZOB flows through a Parshall flume before exiting the ZOB and then drains into La Jara creek.

Daily average precipitation and daily average temperatures were recorded at the Redondo Peak Weather Station (35.8839° N, 106.5536° W, 3231 m above sea level) maintained by the Desert Research Institute's Weather Regional Climate Center. Snow water equivalent (SWE) values were measured at the nearby Quemazon SNOTEL site (35.9167° N, 106.4000° W) located at a similar site elevation (2896 masl) in the Santa Fe National Forest about 5 km



east of Redondo Peak. Cumulative precipitation depths are summed over one water year (from October 1 to September 30 of the next year).

## 2.2 Groundwater well completions

Nested groundwater monitoring wells were drilled to total depths ranging from 6.7 to 47.2 meters below ground surface (mbgs) at each of three locations within the ZOB in June 2016 (Figure 1B and Figure 2). At each location, wells are separated by no more than 2 m from the next well so that the well casings stand in a line. A LiBr tracer was mixed with fluids injected during the drilling and well development process. Br⁻ concentrations were measured in initial samples to ensure that drilling fluids were flushed from monitoring wells prior to analysis of groundwater samples. All screened sections of the well casing have 0.051 cm slotted intervals. All wells were drilled to depth with a diamond impregnated TT coring drill bit at an HQ3 diameter (9.6 cm annulus diameter; Supplemental Table 1). The annuli of all wells were packed with 8-12 CSS sand surrounding screened intervals and the sand packs were sealed with hydrated bentonite pellets.

## 2.3 Water Quality and Age Tracers

Differences in well casing diameter, depth of water column, sampling frequency, and seasonal site accessibility necessitated different sampling collection among wells. Groundwater samples were collected from site 1 wells using a Waterra inertial pump (Waterra USA Inc., Peshatin, WA, USA). Groundwater samples from site 2 wells were collected with a Geotech SS Geosub Pump (Geotech Environmental Equipment, Inc., Denver, CO, USA) except during times when snow accumulated leaving the site inaccessible by vehicle during which times samples were collected with a 42.1 mm stainless-steel bailer (Geotech Environmental Equipment, Inc., Denver, CO, USA). Approximately three borehole volumes were discarded before collecting samples from each well to ensure that formation water was retrieved. Surface water samples of La Jara and ZOB stream water were collected at flume sites as grab samples.

Groundwater and surface water samples were collected in acid washed polypropylene bottles (for cation and trace element analysis) and DI-washed, combusted amber glass bottles (for anion, carbon content, and stable water isotope analysis). Bottles were triple rinsed with sample water prior to sample collection and samples were kept cold until promptly filtered at the University of Arizona. Sample splits for cations and trace elements were filtered through 0.45 µm nylon filters and acidified with trace metal grade nitric acid while all other splits were filtered through 0.7 µm glass fiber filters. Surface water samples were collected biweekly during dry seasons and twice daily at highest and lowest daily discharge using an automatic sampler (Teledyne, ISCO, NE, USA) during spring 2017 snowmelt. Groundwater samples were collected from all wells producing water on a biweekly basis during dry seasons and daily from the shallowest well (Well 2D; total depth 6.7 mbgs) during spring snowmelt using an autosampler (Teledyne ISCO, NE, USA).

All water samples were analyzed for anions (not shown here), cations, dissolved inorganic carbon (DIC), dissolved organic carbon (DOC, not shown here), and stable water isotopes. Cations were measured by inductively coupled plasma mass spectrometry (ICP-MS) at the University of Arizona Laboratory for Emerging Contaminants



(ALEC). Dissolved inorganic carbon was measured with a Shimadzu TOC-VCSH carbon analyzer in ALEC. Stable water isotopes ($\delta^{18}O$ and $\delta^{2}H$) were measured with a DLT-100 Laser Spectrometer at the University of Arizona. Analytical precision ($1\sigma$) for all samples was 0.4 permil for $\delta^{2}H$ and 0.1 permil for $\delta^{18}O$. Stable isotope data from

several previous studies at the JRB-CZO and surrounding locations were also incorporated (Vuataz and Goff, 1986; Longmire et al., 2007; Gustafson, 2008; Broxton et al., 2009; Zapata-Rios et al., 2015b).

Tritium analysis was completed at the University of Arizona Environmental Isotope Laboratory on select groundwater and surface water samples filtered through 0.45 $\mu$m nylon filters. Mean residence time (t) was calculated using the two most common lumped parameter models, the piston flow and exponential models, according to Eq. 1

and 2, respectively (Zuber and Maloszewski, 2000; Manga, 2001; Suckow, 2014; Zapata-Rios et al., 2015b)

$$C = C_0 e^{-\lambda t} \qquad \text{Piston Flow Model} \qquad (1)$$

$$C = \frac{C_0}{1+\lambda t} \qquad \text{Exponential Model} \qquad (2)$$

where C is the measured tritium concentration (reported as tritium units, TU) in groundwater, $C_0$ is the measured TU

in local precipitation, and $\lambda$ is the tritium decay constant (5.576 x $10^{-2}$ year$^{-1}$) based on a tritium half-life of 12.43 yr. Tritium analysis of precipitation in Albuquerque, NM (70 km from the study site) was measured monthly from 1962 to 2005 and are available as part of the Global Network of Isotopes in Precipitation (GNIP) through their Water Isotope System for data analysis, visualization, and Electronic Retrieval (WISER) database. Eastoe et al. (2012) found that volume weighted mean (VWM) tritium concentrations in Albuquerque precipitation have remained stable since the

early 1990s and are similar to local prebomb atmospheric tritium concentrations. Zapata-Rios et al. (2015b) used SWE data from the previously described Quemazon station to calculate a VWM of background tritium (8.6 TU from 1992 to 2005) that is used as $C_0$ in mean residence time calculations herein. The uncertainty of age calculations was computed according to Eq. 3 following Bevington and Robinson (1992), Scanlon (2000), and Zapata-Rios et al. (2015b)


$$\sigma_t = \left[ \left( \frac{dt}{dc_0} \right)^2 \sigma_{c_0}{}^2 + \left( \frac{dt}{dc} \right)^2 \sigma_c{}^2 \right]^{0.5} \qquad \text{Age Uncertainty} \qquad (3)$$

where $\sigma_t$ is the uncertainty of the age calculation, $\sigma_{C_0}$ is the uncertainty of the background tritium concentration, and $\sigma_C$ is the uncertainty in the measurement of tritium in samples in this study.

Groundwater samples from wells 2D and 2C were collected with Hydrasleeve$^{TM}$ (GeoInsight, Las Cruces, NM, USA) passive collector bags in February 2018 for $^{14}$C analysis. Self-sealing bags were deployed in the wells and left to equilibrate for 24 h before sample retrieval. Radiocarbon analysis was completed at the University of Arizona Accelerator Mass Spectrometry (AMS) Laboratory on DIC in groundwater samples. Measured $^{14}$C activities are expressed as percent modern Carbon (pmC) that were calculated as weighted averages of combined machine runs to

reduce overall error and $\delta^{13}C_{DIC}$ measured values are expressed as permil.

Uncorrected radiocarbon ages were computed following Eq. 4 for radioactive decay using a $^{14}$C half-life of 5730 years and an initial $^{14}$C activity ($A_0$) of 100 pmC, according to data from the neighboring Española Basin bound to the west by the Jemez Mountains (Manning, 2009). In order to calculate corrected radiocarbon ages, the $\delta^{13}C_{DIC}$ value of





the recharge water (-16.13 ‰) was first calculated based on an assumed temperature of recharge of 9.6 ºC (the average
temperature of soil water from the ZOB over the last year of consecutive measurements, 2011), pH of 7.51 (the average
pH of soil water from the ZOB over the last year of consecutive measurements, 2011), and a $\delta^{13}C_{CO2}$ of ZOB vegetation
of -24.7 ‰ (unpublished data from Tjasa Kanduc) using equilibrium constants of the carbonate system (Drever, 1997),
alpha values between $CO_{2(g)}$ and $CO_{2(aq)}$ according to Vogel et al. (1970), and alpha values between $HCO_3^-$ and $CO_{2(aq)}$
and between $HCO_3^-$ and $CO_3^{2-}$ according to Mook et al. (1974).

Corrected ages were calculated using the $\delta^{13}C$ mixing model of equations 5 and 6 following Pearson (1965),
Pearson and Hanshaw (1970), and Clark and Fritz (1997) using an assumed $\delta^{13}C_{DIC}$ of calcite of 0 ‰, a calculated
value of $\delta^{13}C_{DIC}$ value of the recharge water, and $A_0$ of 100 pmC.

$$A_{sample} = A_0 e^{-\lambda t} \qquad\qquad \text{Uncorrected Age} \qquad\qquad (4)$$


$$q_{\partial 13C} = \frac{\partial^{13}C_{measured} - \partial^{13}C_{carbonate}}{\partial^{13}C_{recharge} - \partial^{13}C_{carbonate}} \qquad\qquad \delta^{13}C \text{ Fraction from Carbonate Dissolution} \qquad (5)$$

$$A_{sample} = q_{\partial 13C} * A_0 e^{-\lambda t} \qquad\qquad \text{Corrected Age} \qquad\qquad (6)$$


### 2.4 Streamflow and Groundwater Depths

Streamflow measurements were recorded at 15 min intervals from pressure transducers inside the stilling well of
a Parshall flume at the base of the ZOB at 2996 masl and La Jara catchment at 2739 masl (flume locations shown in
Figure 1B). Transducer data were calibrated by depth measurements taken by hand at the time of each sampling event.

All monitoring wells were installed with vibrating wire piezometers (VWP) during the drilling campaign to
provide nearly continuous (15 min) measurements of hydraulic head in each well. However, several piezometers failed
and were replaced with Level Troll 500 (In-Situ Inc., Fort Collins, CO, USA) series pressure transducers (Site 1) and
Rugged Troll 200 (In-Situ Inc., Fort Collins, CO, USA) series pressure transducers (Site 2) in October 2017
(Supplemental Table 1). In this study, time series of groundwater depths are only shown from two wells (Well 1A and
Well 2D) with continuous monitoring via VWPs. Electronic sounder measurements of depth to water were taken
before every groundwater sampling event, converted into depth of water column above VWP, and used to calibrate
VWP and transducer measurements.

### 2.5 Saturated hydraulic conductivity estimates

Nearly continuous monitoring of hydraulic head in groundwater wells enabled individual sampling events to be
treated as rising-head bail-down slug tests (Butler, 2015). Aqtesolv software (Duffield, 2007) was used to curve fit
aquifer response (VWP measurements of depth of water converted into normalized change in water table height
against the logarithm of elapsed time) using the Kansas Geological Survey (KGS) model (Hyder et al., 1994) for three
sampling events in wells 1A and 2D to estimate saturated hydraulic conductivity ($K_{sat}$) values using the Aqtesolv
Automatic Estimation procedure. Early time recovery (first 15 min after disturbance) was missed between time





intervals so the largest normalized head displacement is approximately 0.8 (m m$^{-1}$). Loss of early time data (beginning

$H/H_0$ of 0.8) from Pratt 4-2 slug test data (Butler, 1998; data available from http://www.aqtesolv.com/examples/uncslug1.htm) produced 1.07% error in $K_{sat}$ values. Slug test analysis assumed isotropy.

**2.6 Volumetric water content in soil pedons**

Decagon EC-5 soil moisture sensors measured volumetric water content (VWC) in 6 instrumented soil pedons within the ZOB (Figure 4) ranging from depths of 9.5 to 65 cm below ground surface (cm bgs). VWC was measured at multiple depths in four of the six pedons. VWC was recorded every 10 min.

**2.7 Downhole neutron probe surveys**


Water content profiles with depth were determined from neutron probe (Model 503DR, Campbell Pacific Nuclear, Concord, CA, USA) surveys within the vadose zone in the top 18.3 mbgs in wells 2B and 2C and the total depth of well 3B (12.9 mbgs) over four different events. Raw neutron counts were recorded by the detector using a 32 second interval. Measurements were recorded every 0.3 m and a minimum of three readings were taken at each depth.

Standard counts were measured in an acrylic sleeve before and after measurements of each well and were used to correct for radioactive decay of the Am-241:Be source. Wet and dry calibrations were completed by measuring neutron counts within a 55-gallon drum filled with Viton sand surrounding a PVC casing (same material as well casing). The calibration between neutron counts and water content is generally linear up to water contents of 0.4 (Rempe, 2016), which is greater than the maximum water contents measured here; however, neutron probe counts are sensitive to

changes in bulk density and variable solid phase chemical composition (Gardner et al., 2000; Rempe, 2016). Because of the highly heterogeneous geology and mineralogy in the JRB-CZO ZOB (Moravec et al., In Review), one calibration material (Viton sand) was used for all wells and all depths, which limits the interpretation of neutron probe surveys to comparison over time at the same site.

**2.8 Fracture Traces and Density Estimates**

Downhole images captured with an optical televiewer at site 1 and 2 wells were used to identify fracture traces and calculate fracture density using the Fracture Pattern Quantification (FracPaQ) MATLAB$^{TM}$ toolbox available as open source code via Mathworks$^{TM}$ FileExchange (Healy et al., 2017). Near surface borehole instability required steel

casing from the surface down to approximately 15 mbgs, which necessitated that downhole images, and thus fracture density characterization, started at approximately 15 mbgs. It was not possible to capture downhole images from site 3 wells because the boreholes had to be immediately cased over their entire depth to control the highly pressurized sands encountered during drilling at this site. Because of the variable mineralogy and corresponding color transitions in the boreholes, a node file of fracture locations was needed and was created by tracing fractures onto a new layer

over optical televiewer images from wells 1A and 2A in Adobe Illustrator$^{TM}$. The scalable vector graphics (.SVG) file





of the polyline fracture traces without the underlying downhole image was used as the input file for FracPaQ (Healy et al., 2017), which extracted the *x* and *y* fracture-trace coordinates from the file. Fracture density is defined as the number of fractures per unit area and has the units of m$^{-2}$ (Dershowitz and Herda, 1992). FracPaQ (Healy et al., 2017) calculates fracture density from the fracture segment network as m/2πr$^2$ according to Mauldon et al. (2001) where m

is the number of trace endpoints in circle of radius r by resampling the exposed traces using circular scan windows that eliminate orientation, censoring, and length biases.

## 3    Results

### 3.1 Hydrologic response


Temporal analysis of the climatic parameters (daily precipitation and temperature, Figure 3A, and snow water equivalent (SWE), Figure 3B) that drive streamflow response in ZOB and La Jara surface waters (Figure 3C and Figure 3D, respectively) and groundwater from the fractured tuff (Well 1A, Figure 3E) and shallow perched aquifers (Well 2D, Figure 3F) are used to examine seasonal hydrologic responses. Snow water equivalent peaked at 213 mm

during WY 2017 while annual cumulative precipitation was 673.5 mm for water year 2017. The average temperatures during summer and winter months of WY 2017 were 13.4°C and -4.2 °C, respectively. As temperatures rose above freezing on 3/8/17, the snowpack began to melt and SWE values dropped rapidly between 3/9/17 and 3/20/17. In response to snowmelt, daily averaged discharge of the ZOB flume peaked at 0.00846 m$^3$ s$^{-1}$ on 3/21/17 while La Jara streamflow peaked at 0.142 m$^3$ s$^{-1}$ one day later on 3/22/17. Temperatures dropped below freezing

again between 3/23/17 and 4/5/17 causing ZOB streamflow to freeze producing no measured discharge while La Jara streamflow reached a local minimum on 4/5/17. During freezing temperatures, a second short-lived snowpack accumulated 18 mm of SWE. As temperatures increased above freezing again, maximum flows of 0.158 m$^3$ s$^{-1}$ were reached in La Jara flume on 4/17/17 and a local maximum of 0.00166 m$^3$ s$^{-1}$ was reached in ZOB flume on 4/15/17. While streamflow peaks were greatest in response to spring snowmelt, there were also obvious, smaller peaks in

ZOB and La Jara surface waters following summer monsoons and fall storms. For example, ZOB streamflow peaked at 0.00194 m$^3$ s$^{-1}$ on 7/31/17, 0.00194 m$^3$ s$^{-1}$ on 8/4/17, and 0.00245 m$^3$ s$^{-1}$ on 9/30/17 while La Jara streamflow peaked several times, most notably at 0.116 m$^3$ s$^{-1}$ on 7/31/17, 0.0926 m$^3$ s$^{-1}$ on 9/27/17, and 0.0890 m$^3$ s$^{-1}$ on 9/30/17. Both snowmelt and rainfall events influenced La Jara creek and ZOB streamflow leading to increases above baseflow (Figure 3C and D, respectively). However, the major driver of streamflow and groundwater depth

response was spring snowmelt.

The depth of water in wells 1A and 2D (Figure 3D and F, respectively) also responded to spring snowmelt, summer monsoons, and fall rainfall. Well 1A water depths peaked on 3/22/17 just two days after SWE values dropped to zero and on the same day that La Jara streamflow reached a local maximum. As temperatures froze again and a second snowpack developed, well 1A depth of water receded until 4/6/17 after which it quickly rose to a

second local maximum on 4/11/17 four days before ZOB streamflow reached a second max and six days before La Jara streamflow reached its second peak. Well 1A water depths also peaked sharply on 10/2/17 in response to fall


precipitation. Conversely, well 1A water depths reached one gradual peak during NAM season on 8/16/17, despite several smaller La Jara streamflow peaks in response to summer monsoon storms.

Water depths in well 2D (Figure 3F) did not have pronounced, sharp maxima and minima in response to snowmelt dynamics like those seen in surface waters and site 1 groundwater. Instead, water depths in well 2D continued rising, with slight changes in rate on 3/21/17 and 4/4/17, while temperatures dropped below freezing again and a second snowpack accumulated and melted before well 2D water depths peaked on 4/22/17. Well 2D water depths did not respond to summer monsoons and, instead, continuously decreased from their spring snowmelt peak at 4/22/17 until reaching a minimum after summer monsoons on 10/4/17. However, well 2D gradually

increased in response to fall storms until reaching peak water depths on 10/25/17. It appears that water slowly infiltrated into the perched aquifer and recharged the near surface groundwater store.

    Volumetric water content (VWC) in ZOB soils from six pedons ranging from 9.5 to 65 cm bgs also varied seasonally with generally higher VWC during spring snowmelt, lower VWC at the onset of NAM season, and intermediate VWC during fall storms (Figure 4). VWC changes across the water year at the shallowest depth in

pedons 4 and 3, located in the western area of the ZOB nearest site 2 wells, were small (0.20 $m^3$ $m^{-3}$ range in pedon 4 and 0.16 $m^3$ $m^{-3}$ range in pedon 3), while VWC at greater depth in pedon 3 increased drastically and remained elevated during spring snowmelt (peaked at 0.4175 $m^3$ $m^{-3}$ and 0.4186 $m^3$ $m^{-3}$ for 65 and 30 cm bgs, respectively) and fall storms (peaked at 0.4116 $m^3$ $m^{-3}$ and 0.3109 $m^3$ $m^{-3}$ for 65 cm and 30 cm bgs, respectively). Changes in VWC at pedons 5 and 2, located in the convergent zone of the ZOB nearest site 3 wells, were most pronounced

during spring snowmelt. Again, increased VWC in pedon 2 persisted longer at depth than at the near surface. In contrast, soils in the eastern area of the ZOB near site 1 wells at pedons 6 and 1 did not remain wet for extended periods, but rather increases in VWC of pedons 6 and 1 were flashy while decreased VWC during NAM season was sustained.

    Estimates of saturated hydraulic conductivity of wells 1A and 2D were computed to explore differences in

hydrologic properties of the two aquifers following three sampling events that served as bail-down slug out tests. Averaging $K_{sat}$ from the three events produced a mean for well 2D of 7.22 x $10^{-3}$ m day$^{-1}$ and 1.22 x $10^{-4}$ m day$^{-1}$ for well 1A (Table 1), despite fracture density of deeper site 2 wells estimated to be approximately five times less than fracture density of site 1 wells (Figure 5). However, it is important to note that fracture density could not be calculated across the water table depths of well 2D or site 1 wells because of surface instability and presence of

water inhibiting downhole televiewer images.

### 3.2 Water routing in unsaturated zone

    Soil moisture content in two site 2 boreholes (Wells 2B and 2C) and one site 3 borehole (Well 3B) were estimated

from a neutron probe soil moisture gauge that was run downhole on four dates. Due to the textural shifts and complexity of mineral composition as a function of depth at each site and across sites (Moravec et al., In Review),





water content estimates are used to qualitatively examine changes in water content with depth and over time within each respective borehole.

Profiles of water content with depth below ground surface in borehole 2B (Figure 6, left), reach a local
maximum water content of approximately 0.32 cm$^3$ cm$^{-3}$ at 1 mbgs during all events. At increased depth, the water content recedes to an overall minimum of 0.15 cm$^3$ cm$^{-3}$ at 2.4 mbgs in the June, August, and February measurements while the October measurement recedes to only 0.23 cm$^3$ cm$^{-3}$ at 1.8 mbgs before spiking to 0.36 cm$^3$ cm$^{-3}$ at 2.4 mbgs and receding slightly to 0.35 cm$^3$ cm$^{-3}$ at 4 mbgs below which depth changes in water content are consistent across all time series.

In borehole 2C (Figure 6, center), just 2 m away from borehole 2B, differences in water content profiles are also seen over time. In October, water content increases from the surface until reaching an overall maximum of 0.4 cm$^3$ cm$^{-3}$ at 3.7 mbgs whereas profiles from the other three events peak at 0.29 cm$^3$ cm$^{-3}$ at only 0.6 mbgs, recede to 0.08 cm$^3$ cm$^{-3}$ at 1.2 mbgs and increase gradually to 0.12 cm$^3$ cm$^{-3}$ at 3.4 mbgs and drastically jump up to a range of water content from 0.34 to 0.39 cm$^3$ cm$^{-3}$ (July to Feb, respectively) at 4 mbgs. From 4 to 15.5 mbgs, the water
content profiles for all events are relatively consistent between 0.28 cm$^3$ cm$^{-3}$ and 0.30 cm$^3$ cm$^{-3}$ (June and August identical, October 0.1 greater than summer profiles, and Feb 0.1 greater than October).

Finally, water content profiles with depth in borehole 3B (Figure 6, right) are nearly identical across all measurement events and show three peaks in water content of 0.36 cm$^3$ cm$^{-3}$, 0.33 cm$^3$ cm$^{-3}$, and 0.25 cm$^3$ cm$^{-3}$ at 1.2, 8.5, and 11.6 mbgs, respectively.


### 3.3 Contribution of distinct groundwater stores to streamflow

In a volcanic setting such as the Valles Caldera, silicate mineral weathering is the primary driver of stream water chemical fluxes (McIntosh et al., 2017) and larger concentrations of base cations have been found in waters with
longer flow paths as mineral dissolution fluxes increase with increasing water transit times (Zapata-Rios et al., 2015b). Quantitative mineralogy of cores collected during the June 2016 drilling campaign in the ZOB (Moravec et al., In Review) found that quartz, potassium feldspar, plagioclase, volcanic glass, and smaller percentages of mica are the primary minerals ubiquitous in site 1 and 2 cores. Smectite, iron oxides, illite, and magnesite, as well as diagenetically altered minerals like Ca-zeolites (clinoptilolite and mordenite), are present in smaller percentages within the top 15
mbgs of site 1, along with some 2:1 clays from 15 to 17 mbgs and 20 to 26 mbgs. Ca-zeolites, smectite, illite, iron oxides, and trace talc and tremolite are also found throughout site 2 cores, as well as secondary minerals like calcite and illite in the top 15 mbgs. Greater percentages of 2:1 clays are also found throughout site 2 cores, especially from 12 to 16 mbgs and 22 to 30 mbgs. Previous analysis of saturation indices of ZOB groundwater found that well 2D shallow groundwater was saturated with respect to calcite (Olshanksy et al., 2018) while previous work by Zapata-
Rios et al. (2015b) found that springs across the JRB-CZO were undersaturated with respect to calcite, albite, and sanidine; therefore, interaction with those minerals is expected to influence groundwater and surface chemistry in the ZOB and La Jara catchment.



The primary groundwater cations from each monitoring well are $Ca^{2+}$, $Na^+$, and $Mg^{2+}$ (Table 2); however, the percentages (in terms of µeq/L) of each cation differ between sites (Sites 1 and 2). Major ion concentrations also differ

with depth between site 2 wells while site 1 groundwaters (Wells 1A and 1B) are geochemically similar to one another (Supplemental Fig. 1). These differences in cation concentration are used in further analysis to distinguish streamflow sources.

Temporal analysis of major cation concentrations in groundwater and surface water over WY 2017 (Figure 7) again shows clear separation of groundwater concentrations between the two sites. $Ca^{2+}$ and DIC concentrations are

highest in the perched aquifer (Well 2D), where calcite is known to be present, and all site 2 groundwater $Ca^{2+}$ and DIC concentrations are considerably greater than surface and site 1 waters year-round (Figure 7). Furthermore, $Ca^{2+}$ and DIC concentrations are most variable in wells 2D and 2C, which change together in time, suggesting a connection between the two water stores. Finally, both $Ca^{2+}$ and DIC concentrations of shallow groundwater increase simultaneously, which is consistent with calcite dissolution, at the time that La Jara streamflow increases

above baseflow during snowmelt. In contrast, $Ca^{2+}$ and DIC concentrations of La Jara surface waters do not change markedly as streamflow increases while $Ca^{2+}$ and DIC concentrations of ZOB surface waters decrease slightly during this time. $Mg^{2+}$ concentrations are greatest in the deepest well (Well 2A) and decrease with decreasing depth at site 2, are lower still in site 1 groundwater, and lowest in La Jara stream water (Table 2). $Na^+$ concentrations of the perched aquifer increase steadily at the onset of snowmelt while La Jara and ZOB surface water $Na^+$

concentrations decrease slightly (Figure 7). Unfortunately, the lack of site 1 groundwater data during snowmelt makes it impossible to determine if the variability of site 1 groundwater chemistry resembles that of surface waters; however, surface water concentrations of major ions are generally closer in magnitude to those of site 1 groundwaters throughout the remainder of the water year.

Comparison of $Ca^{2+}/Mg^{2+}$ molar ratios with $Na^+$ concentrations (Figure 8A) and DIC concentrations (Figure

8B), which can differentiate between weathering of $Ca^{2+}$ rich and $Mg^{2+}$ rich silicate minerals, also show distinct groupings of groundwater between sites and depths. The $Ca^{2+}/Mg^{2+}$ molar ratios of the perched aquifer are greater than those of the deeper waters. The $Ca^{2+}/Mg^{2+}$ molar ratios of surface waters overlap in space with those of deeper groundwater. The DIC concentrations clearly differentiate between sites 1 and 2 and the surface waters plot with similar DIC concentrations as site 1 groundwater, which indicates that deeper groundwater from site 1 is more

representative of streamflow in La Jara catchment.

### 3.4 Mix of old and young snowmelt dominated waters

Stable water isotopes of groundwaters and surface waters plot together in space and plot closer to the lower

isotope values of snow than to those of summer precipitation (Figure 9) indicating that snowmelt is the dominant source of recharge to groundwaters and surface waters in the ZOB and La Jara catchment. The majority of samples plot along the local meteoric water line (LMWL; from Broxton et al. (2009)), showing consistency in stable water isotopes over time. However, several surface waters and a few samples of deep groundwater from site 1 wells and



shallow groundwater from well 2D have higher $\delta^{18}O$ and $\delta^2H$ values that plot to the right of the LMWL, along an
enrichment line with slope of 3.4 (Figure 9).

Tritium was detected in groundwater from each sampled well, which indicates that there is a component of modern recharge to all groundwater stores (Manning, 2009). Tritium content from wells analyzed in June 2017 and February 2018 are within two standard errors of one another indicating little difference between the tritium content of groundwater stores between the summer dry season and the winter. The highest tritium content (4.4 TU in February
2018) and therefore shortest residence time waters (12 and 17 years according to piston flow and exponential models, respectively) are those of the perched aquifer while the lowest tritium content (0.7 TU in February 2018) and longest residence times (45 and 202 years, according to piston flow and exponential models, respectively) are from wells 2B and 2C (Table 3). As expected, there is more tritium present in the shallowest groundwater compared to deeper waters from site 2 wells; however, the deepest site 2 groundwater from well 2A has more tritium than the shallower wells 2B
and 2C. Differences in tritium content (Table 3) across similar depths from sites 1 and 2 (Figure 2) indicate the presence of separate groundwater stores of water within the ZOB.

Radiocarbon age calculations were computed for the shallowest groundwater of the perched aquifer from well 2D and groundwater beneath the perched aquifer from well 2C based on $^{14}C$ activity and $\delta^{13}C$ of DIC of the two wells. The $^{14}C$ activity of DIC from well 2D was 75.34 ± 0.19 pmC while the $\delta^{13}C_{DIC}$ was -13.1 ‰, which corresponds to a
corrected radiocarbon age of 621 years. The $^{14}C$ activity of DIC from well 2C was 60.02 ± 0.17 pmC and the $\delta^{13}C_{DIC}$ was -12.4 ‰, which corresponds to a corrected radiocarbon age of 2050 years (Table 3). As expected, there is less modern $^{14}C$-DIC in the deeper groundwater (Well 2C) indicating longer residence times at greater depth.

**4    Discussion**

This study seeks to understand the seasonal hydrologic response of groundwater as a function of depth below ground surface at sites 1 and 2 and to explore how CZ architecture influences seasonal groundwater contribution to streamflow in La Jara catchment. Contrasting CZ architecture and lithology in sites 1 and 2, along with time series of
shallow and deep groundwater from both sites and surface water from the catchment outlet, enable us to decipher geochemical signatures of deep groundwater from a highly fractured aquifer and shallow groundwater from a perched aquifer. The following sections discuss the dynamics of seasonal hydrologic response, water routing, recharge and water residence times, major ion chemistry, and CZ architecture to investigate streamflow contributions over time.

**4.1 Hydrologic response**

Site 1 wells are situated in highly fractured welded tuff with maximum fracture density of approximately 5000 $m^{-2}$. This contrasts with highly weathered volcanic breccia in the top 15 mbgs at site 2 wells that corresponds with a caldera collapse breccia deposit overlying more consolidated, less fractured welded tuff with maximum fracture
density of approximately 1000 $m^{-2}$ at depth (Figure 5). We hypothesize that the differences in subsurface structure, presence of a confining layer, and greater fracture density of site 1 wells compared to site 2 wells (Figure 5)



influences the different hydrologic response of the two sites (Figure 3). Well 1A responds rapidly to snowmelt, reaching its first peak on the same day as La Jara streamflow's first peak, and responds gradually to summer monsoon events. Well 2D has a muted response to snowmelt with the first inflection point occurring on the same day as the first ZOB discharge peak while the first well 2D water table maximum occurs 45 days after the onset of snowmelt and does not respond to summer monsoon events at all.

Mean $K_{sat}$ values from both wells are lower than $K_{sat}$ estimates of the same Tshirege member of Bandelier Tuff from nearby Los Alamos National Lab that range from 7.6 x $10^{-2}$ to 1.12 m day$^{-1}$ (Rogers and Gallaher, 1995; Smyth and Sharp, 2006) and 9.3 x $10^{-1}$ to 1.6 x $10^{1}$ m day$^{-1}$ (Kearl et al., 1990; Smyth and Sharp, 2006). This may be, in part, because slug tests provide localized estimates of the hydraulic conductivity directly surrounding the screened interval of the well in contrast to pumping tests that provide larger scale volumetric averages of hydraulic properties. Butler (2015) notes that hydraulic conductivity estimates from slug tests should be viewed as lower bounds of the conductivity in the vicinity of the well. Furthermore, the degree of welding and presence of alteration may account for the discrepancies in conductivity estimates as Rogers and Gallaher (1995) noted that the degree of welding of the Bandelier Tuff was greatest closer to the Valles Caldera, the tuff's volcanic source and site of this study.

Surprisingly, the mean $K_{sat}$ of the perched aquifer (Well 2D) is more than one order of magnitude greater than that of the deep well 1A (7.22 x $10^{-3}$ m day$^{-1}$ and 1.22 x $10^{-4}$ m day$^{-1}$, respectively, Table 1). Many more fractures are present at site 1 relative to site 2 (Figure 5A), which generates a fracture density approximately 5 times greater at site 1 wells than site 2 wells (Figure 5B). Unfortunately, it is not possible to directly compare the fracture density across the screened intervals from which hydraulic conductivity estimates were made because downhole images could not be captured within either well at those depths, but we expect that the general trend of dense fractures at site 1 and few fractures at site 2 would persist. The lower mean $K_{sat}$ of well 1A could suggest that fractures may be backfilled by mineral precipitates and weathering rinds decreasing the ability of fractures to act as preferential flowpaths. It is noteworthy, however, that such a mechanism is not clearly supported by the downhole images or core samples; these images and cores do not show obvious evidence of backfilled fractures.

Despite unexpected $K_{sat}$ differences between the two wells, the very similar shape and timing of well 1A's hydrograph compared to those of La Jara stream and ZOB surface water (Figure 3) is a function of pressure propagation and indicates hydraulic connection between the fractured aquifer and streamflow and (Sophocleous, 1991a; Welch et al., 2013). The rapid pressure pulse transfer between site 1 groundwater and the stream suggests that the fractured welded tuff aquifer has low specific storage and high transmissivity. Worthington (2015) noted much more rapid changes in head in low storage fractured bedrock aquifers compared to granular aquifers and Sophocleous (1991a) found that the hydraulic diffusivity is the major control of the extent and speed of pressure pulse propagation.

The rapid response of the welded tuff aquifer (Well 1A) contrasts with the muted response of the perched aquifer (Well 2D), which does not show evidence of pressure pulse propagation between shallow groundwater and the ZOB and La Jara surface waters. The slower response of groundwater levels in well 2D suggests that the perched aquifer is not directly hydraulically connected to the stream likely because of the confining layer separating it from



deeper groundwater and the significantly lower fracture density of site 2 wells (Figure 5). The comparison of site 1

and 2 groundwater hydrographs to surface water hydrographs bears a striking resemblance to the juxtaposition of

stream-flood wave propagation in monitoring wells drilled into buried river channels (similar to well 1A) and the

absence of pressure pulses in monitoring wells not associated with buried channels (similar to well 2D) seen in the

Great Bend alluvial aquifer in Kansas (Sophocleous, 1991a). Furthermore, the cubic shape of the rising water table

in the perched aquifer, as well as the rapid rise after inflection point in well 2D is indicative of groundwater recharge

(Sophocleous, 1988, 1991a, 1991b). Further analysis of the rates of increase before and after the well 2D hydrograph

inflection point can be found in Olshansky et al. (2018).

Hydrologic response to incoming precipitation exhibits strong dependence on season, as indicated by differences

following spring snowmelt, summer monsoons, and fall precipitation. Specifically, well 1A groundwater responds

rapidly to spring snowmelt (water table peaks 35.4 mbgs on 3/22/17) and fall precipitation (water table peaks 35.4

mbgs on 10/2/17); however, the welded tuff aquifer's response to summer monsoon rain is smaller and much more

gradual (peaking at 36.4 mbgs on 8/16/17), suggesting that spring snowmelt and fall precipitation induce a different

hydrologic flow regime than that of summer monsoons. Different hydrologic flow regimes across seasons also exist

in the perched aquifer (Well 2D). While all changes in the perched aquifer water table are gradual, spring snowmelt

and fall precipitation produce water table peaks of 2.3 mbgs on 4/22/17 and 2.7 mbgs on 10/25/17 while the perched

aquifer water table steadily decreased during summer monsoons indicating no water table changes induced by summer

storms.

The less pronounced hydrologic response to summer monsoons, compared to snowmelt and fall rain, in shallow

groundwater of site 2 and deeper groundwater at site 1 is likely a function of drier antecedent soil moisture

conditions at the onset of the NAM season, as indicated by decreased VWC in shallow soils from 6 instrumented

pedons in the ZOB immediately before monsoon storms began compared to wetter soils during spring snowmelt and

fall storms (Figure 4). This agrees with previous studies in the VCNP, which found that soil moisture was lowest in

early summer after soil moisture from snowmelt had receded and it increased after the arrival of monsoon storms

(Vivoni et al., 2008; Molotch et al., 2009). Furthermore, Zapata-Rios et al. (2015a) found that NDVI in the VCNP

increased during the NAM season suggesting that precipitation was partitioned to plant use during monsoon rains

and was not available to recharge groundwater stores. Furthermore, smaller, sporadic precipitation during summer

monsoon storms and increased evapotranspiration due to higher temperatures (Figure 3A) and increased plant use

create a wetting and drying effect in shallow soils that can be seen as small fluctuations in VWC (Figure 4). This

effect likely inhibits infiltration of water into the subsurface and agrees with Langston et al. (2015)'s  model of

unsaturated zone flow in two seasonally snow covered hillslopes in Colorado which found that episodic recharge

inhibited fluid flow down to the water table because of the need for shallow soil to re-wet after each precipitation

event.

**4.2 Water routing in unsaturated zone**





Precipitation inputs differ before each neutron probe measurement (2.54 mm one day prior to July
measurement, 2.03 mm one day prior to August measurement, 5.08 mm six days prior and 0.25 mm one day prior to
October measurement, and no precipitation over 106 days prior to the February measurement; Figure 6 top);
however, water content is nearly identical in the top 4 mbgs in three of the four profiles for site 2 wells and in all site
3 data collection events (Figure 6 bottom). We hypothesize that the response seen at site 2 in the October survey is a
function of both, the slightly larger precipitation depth prior to this survey and the wetter shallow soil conditions
preceding the survey as compared to the conditions preceding the other three surveys. Higher frequency sampling
during wet season are needed to determine the impact of precipitation depth and the potential for precipitation
thresholds to induce vertical flow. Perhaps temporal changes in water content with depth were missed because of the
sporadic timing of neutron probe surveys due to the arduous transportation and permitting issues involved with the
use of this instrument. While it does not appear that water content profiles with depth captured progressive
enrichments in rock moisture as seen by Salve et al. (2012) and Rempe and Dietrich (2018), they do indicate that the
minimum dry season rock moisture storage is consistent across dry seasons, suggest differences in water routing,
and identify lithologic discontinuities in the subsurface.

Neutron probe surveys show small shifts in water content with depth that are likely associated with small scale
heterogeneities in bulk density created by the lithologic discontinuities in volcanic collapse breccia deposits,
variable degrees of ash consolidation, welding, and secondary mineral precipitation, which are evident in
quantitative mineralogical analyses (Moravec et al., In Review). For instance, well logs showed that a thin layer of
coarse gravel-like material underlies the soil, which is expected to drain quickly (high permeability) and retain little
water (low porosity). At the depth corresponding to that gravel-like layer, water content across all times recedes
quickly (Figure 6, borehole 2B) or remains constant (borehole 2C) before increasing rapidly just above the water
table of the perched aquifer. Salve et al. (2012) also found that moisture content variation from neutron probe
surveys in weathered argillite were strongly linked to changes in material properties with depth, which suggested
different flow processes through the unsaturated zone. Here, lenses of increased water content (like that from 9 to 11
mbgs in well 2B) above layers of relatively lower water content (like that of 12 to 13.5 mbgs in well 2B) are
indicative of subsurface lateral flow through more saturated, more conductive media that can be seen in Figure 6.

Evidence of vertical infiltration is also seen in the site 2 wells. The marked change in shape (increased water
content from 1 to 4 mbgs) of the October water content profile suggests vertical infiltration and subsequent recharge
to the perched aquifer. Analysis of well 2D hydrograph (Figure 3F) confirms that this October neutron probe survey
was completed while the perched aquifer water table was rising. Despite similar perched aquifer water table depths
for all four surveys (2.7 mbgs on 6/27/17, 2.9 mbgs on 8/15/17, 2.9 mbgs on 10/12/17, and 3.2 mbgs on 2/6/18), the
October survey was the only survey that corresponds with a rising rather than receding water table (Figure 3F) and is
the only water content profile that captured vertical infiltration of recharge to the perched aquifer. Furthermore, the
wet October profile returns to dry conditions within the top 4 m in February and February water content beneath 4 m
exceeds that of previous surveys, which suggests that water drains vertically at depths greater than 4 mbgs in
February.



Geologic maps of the Valles Caldera (Goff et al., 2011) indicate that a blind fault bisects the ZOB and it appears that site 3 wells were drilled immediately next to (possibly within) the fault zone, which coincides with the convergent outlet of the ZOB (Figure 1). This is likely why site 3 wells did not produce water in the time period of this study (Figure 2). Furthermore, water content is nearly identical across all four measurement events in borehole 3B (Figure 6). Despite being located in a convergent zone subjected to seasonal wetland saturated conditions at the

land surface, neutron probe surveys do not indicate that water infiltrates vertically in site 3. Rather, data suggest that water moves laterally in the subsurface as indicated by the three lenses of increased water content seen in Figure 6. This is further supported by high clay content (up to 50%) observed below 1.5 m depth at site 3 (Moravec et al. In Review), which likely impedes vertical infiltration and induces lateral flow in the subsurface.

**4.3 Contribution of distinct groundwater stores to streamflow**

In the same headwater catchment in the JRB-CZO, Olshansky et al. (2018) observed temporal changes in major ion concentrations of soil, surface, and ground water during spring snowmelt 2017 and found positive Si concentration pulses during the falling limb of surface water hydrographs that were hypothesized to result from increasing

groundwater contributions during receding surface flows because surface water Si concentrations were similar to those of shallow groundwater (Well 2D). However, surface water concentrations of other major ions ($Ca^{2+}$, $Na^+$, $Mg^{2+}$, Mn, and $SO_4^{2-}$) did not coincidentally rise despite higher shallow groundwater concentrations of those ions compared to Si, which suggests, instead, a deeper source of groundwater to La Jara streams. Herein, we present further evidence that the deep groundwater source to La Jara and ZOB surface water is the deep welded tuff aquifer found throughout

the greater La Jara catchment and represented at site 1 wells.

High $Ca^{2+}/Mg^{2+}$ molar ratios of shallow groundwater from the perched aquifer (Well 2D) are a function of calcite dissolution, which is only present in the shallow subsurface at site 2 and leads to increased $Ca^{2+}$ concentrations but does not impact $Mg^{2+}$ concentrations. The clear separation of $Ca^{2+}/Mg^{2+}$ molar ratios in shallow (Well 2D) and deep groundwater (Wells 1A, 1B, 2A, 2B, and 2C) and the overlap of surface water $Ca^{2+}/Mg^{2+}$ molar ratios with those of

deeper groundwater suggests that the perched aquifer does not contribute substantial volumes to surface flow, but instead deep groundwater is the dominant source of streamflow.

The dissolution of calcite in the perched aquifer also leads to higher DIC concentrations in waters in the ZOB (Table 2; Appelo and Postma, 2005). Higher DIC concentrations in all site 2 waters is consistent with vertical connection between site 2 wells that, in turn, suggests the confining layer beneath the perched aquifer acts as more of

an aquitard than aquiclude. The $Ca^{2+}/Mg^{2+}$ molar ratios and DIC concentrations of surface waters are very similar to those of site 1 groundwaters, which again indicates that deep groundwater from site 1 wells is more representative of groundwaters that contribute to La Jara stream. The welded tuff rock type (Figure 1) of site 1 wells is also more representative of the geology, by volume, throughout the greater La Jara catchment.

Temporal analysis of major ion chemistry indicates that deep groundwater from fractured tuff (Site 1) sustains

stream baseflow as streamflow concentrations and trends in concentrations over time are consistent with site 1 groundwater concentrations. In contrast, pronounced changes in shallow site 2 groundwater (Wells 2D and 2C) major





ion chemistry are not reflected in streamflow concentrations over time, which suggests that shallow groundwater represents only a small volumetric contribution to streamflow. Furthermore, recent work by Olshansky et al. (2018) found that soil water was an important contributor to surface water during spring snowmelt 2017 and may explain the subtle trends in $Ca^{2+}$ and DIC concentrations of surface waters, particularly ZOB surface water which was correlated with soil $P_{CO2}$ concentrations, at the onset of snowmelt.

In summary, we propose a schematic model (Figure 10) to conceptualize the details of hydrologic structure and hydrologic function at two contrasting hillslopes within the ZOB (Sites 1 and 2). Site 2 has multiple, separate stores of groundwater across depth that are distinct from each other and distinct from deep groundwater stores at site 1. All groundwater is recharged via snowmelt and seasonal differences in hydrologic response to precipitation inputs exist at both sites with less pronounced response to summer monsoons at both sites linked to drier antecedent shallow soil moisture at the onset of NAM season. There is evidence of vertical infiltration and subsurface lateral flow at site 2 and a mix of young and older waters, which are expected to persist across all groundwater stores. The fracture density at site 1 is approximately 5 times greater than at site 2 and the CZ structure and architecture of site 1 is most representative of the greater La Jara catchment. Deep groundwater from the fractured aquifer at site 1 is hydrologically connected to streamflow and site 1 deep groundwater chemistry is most representative of water contributing to streamflow while the distinct chemical signature of shallow groundwater from site 2 is not seen in streamflow.

## 4.4 Mix of old and young snowmelt dominated waters

The grouping of most stable water isotopes in Figure 9 indicates that snowmelt is the dominant source of recharge to all groundwater stores in this study as the stable water isotope signatures of surface and groundwater plot much closer to the volume weighted mean (VWM) of snow (Gustafson, 2008) than to the VWM of summer precipitation (Zapata-Rios et al., 2015b). Furthermore, the detection of tritium in groundwater from each sampled well indicates a component of modern recharge to each groundwater store, which agrees with previous work that found springs surrounding Redondo Peak are composed of modern water (Zapata-Rios et al., 2015b). The presence of tritium in groundwater suggests that snowmelt slowly infiltrates into all groundwater stores (Figure 10). However, radiocarbon analysis of wells 2D and 2C also indicates the presence of much older waters (corrected ages of 621 and 2050, respectively) in these shallow aquifers (Figure 10). The detection of tritium and less than 100 pmC in these wells suggests a mixture of old and young waters (Bethke and Johnson, 2008; Jasechko et al., 2017); that is expected to persist in each groundwater store.

Decreasing tritium content with depth from 2D to 2C to 2B (Table 3) agrees with previous studies that have suggested residence times increase with depth (Zapata-Rios et al., 2015b). This trend along with the distinct $Ca^{2+}/Mg^{2+}$ molar ratios and DIC concentrations from the presence of shallow calcite deposits in site 2 wells is consistent with a vertical connection between the three shallowest site 2 wells. However, increased tritium content in well 2A (closer to that of site 1 than other site 2 wells) suggests that 2A is not vertically connected to the above site 2 wells and, instead, is likely laterally connected to younger waters upgradient.



The few samples that plot to the right of the meteoric water line may be related to evaporation and/or the presence
of geothermal waters. The slope of the evaporation trend (m=3.4) is consistent with the slope of evaporation trends
seen in arid environments like southern Arizona and northern Mexico (Gray, 2018; Zamora, 2018). The timing of the
lower surface water isotope values is indicative of evaporative enrichment; however, the sporadic timing during
surface water baseflow of the lower isotope values for all groundwaters is surprising (Supplemental Fig. 2).

The second possible explanation of this enrichment trend is the presence of geothermal waters in the ZOB. The
observed enrichment trend is consistent with that seen by Vautaz and Goff (1986) from geothermal waters sampled
from lower elevations in the Valles Caldera and surrounding area. However, analysis of other geothermal water
indicators ($pCO_2$ and temperature) does not suggest the presence of geothermal waters in the ZOB. Furthermore, $SO_4^{2-}$
and $Cl^-$ concentrations (data not shown) are several orders of magnitude lower than those of known geothermal waters
from Vuataz and Goff (1986).

## 5   Conclusions

There are multiple separate stores of groundwater in the subsurface of the ZOB at the JRB-CZO. Major ion
chemistry show that these aquifers are chemically distinct while analysis of stable water isotopes indicates that
snowmelt is the dominant source of recharge to all groundwater stores. Furthermore, seasonal differences in
hydrologic response to periodic precipitation inputs, including a less pronounced response to summer monsoons, are
seen in groundwaters within the ZOB. Tritium concentrations and radiocarbon analysis indicate that groundwaters are
a mix of modern recharge with older waters. The path and timing of water moving through the subsurface is influenced
by subsurface structure, which is exemplified by variable water contents with depth as a function of within-catchment
lateral and vertical lithologic variation. The perched aquifer at site 2 resides in a disconnected collapse breccia deposit
of different geology from the other wells, is geochemically different from La Jara stream water, and does not respond
to the pressure pulse associated with increasing streamflow and rising water table in the deep site 1 aquifer, which
suggests that the perched aquifer does not contribute significantly to streamflow.

We conclude that the fractured rock site 1 architecture is most representative of the CZ volume that dominantly
contributes to La Jara streamflow across the water year. The similarity in shape and timing of well 1A and surface
water hydrographs results from pressure pulse propagation through the transmissive, low storage fractured aquifer.
Furthermore, the clear similarities in major ion chemistry confirm the connection between La Jara stream and site 1
groundwater. Deep groundwater from site 1 wells appear to be more chemically-representative of waters that
contribute to La Jara stream and more representative of the structure (geology, fractured aquifer, deep groundwater,
longer mean residence times) and function (hydrologic response, solute fluxes, and water routing) of the CZ in the
greater La Jara catchment. The dominant contribution of deep groundwater to surface flows and the hydraulic
connection between the fractured aquifer and streamflow may suggest that deep groundwater stores in fractured
aquifers are sensitive to changes in climatic drivers of streamflow like shifts in precipitation magnitude and timing as
predicted in the southwestern United States.

## Acknowledgements





This research was funded by the U.S. National Science Foundation grants EAR-0724958 and EAR-1331408, which support the Catalina-Jemez River Basin Critical Zone Observatories. Additional funding was provided by a Geological Society of America Student Research Grant awarded to A. White. We are grateful to Mary Kay Amistadi, Tim Corley, Nicole Vicenti, Adam Weber, Mark Losleben, and Adam Killebrew for their assistance with chemical

analyses and field work.

**Author Contribution**

AW performed the data analysis, developed the conceptual framework, and wrote the manuscript with assistance from JM, TF, TM, JC, YO, and BM. AW collected and processed samples with help from BM, AS, and BP. JM and

JC supervised the project.



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




**Figures:**

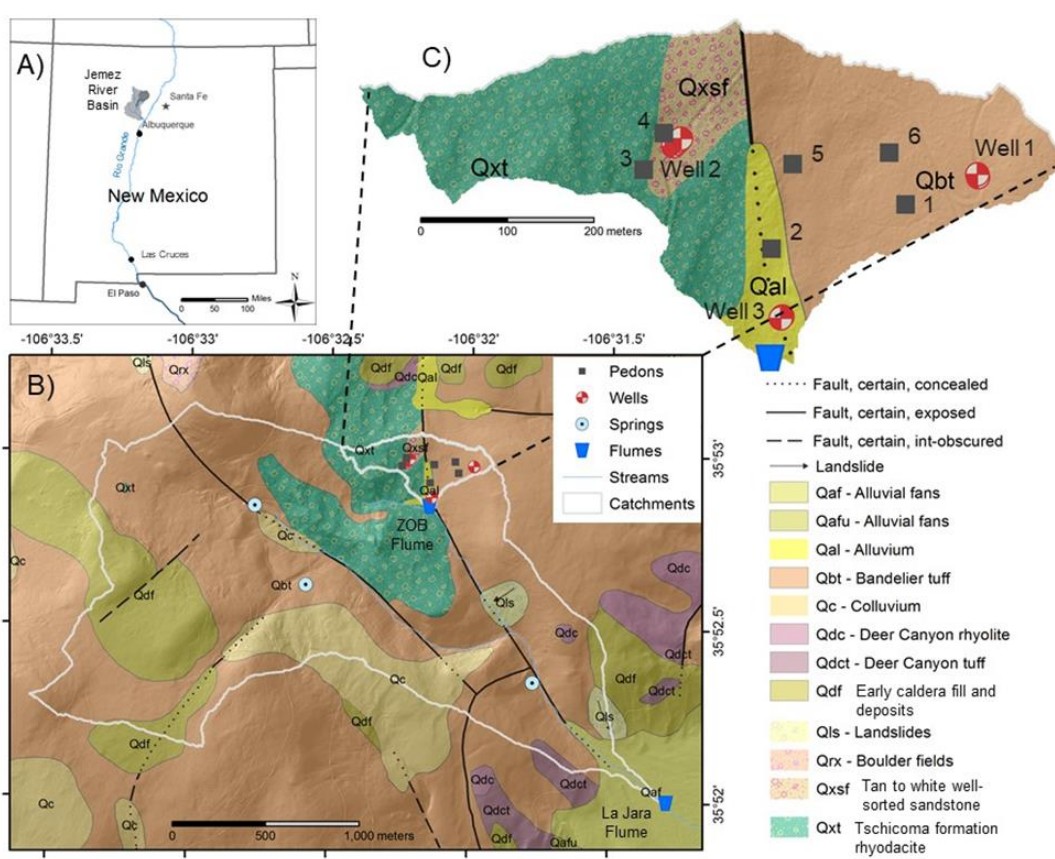

**Figure 1: A) The Jemez River Basin Critical Zone Observatory (JRB-CZO) is located on the Valles Caldera National Preserve in northern New Mexico north of Albuquerque. B) Notice the geologic complexity of La Jara catchment and the Zero Order Basin (ZOB) and the location of surface water flumes and springs. C) Note the different geology of the three well sites, the location of the soil pedons in relation to the wells, and the fault at the site 3 well.**





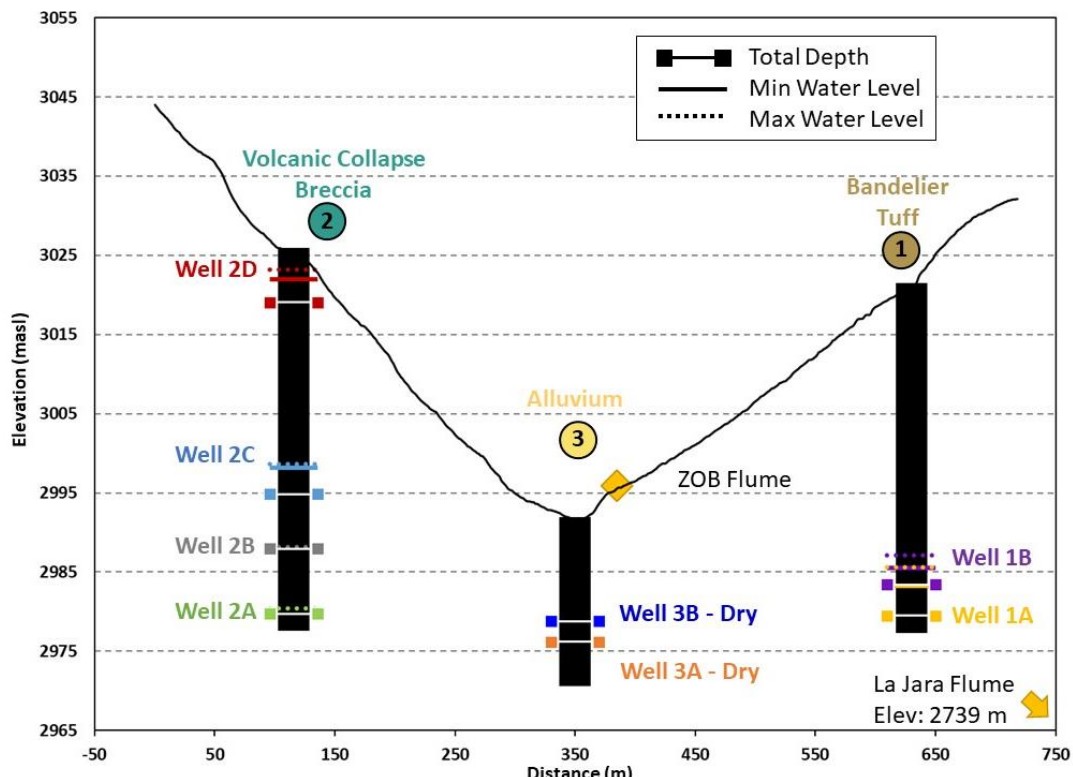

**Figure 2: Profile of the nested groundwater wells within the ZOB showing surface topography, depth of wells, and seasonal changes in water column heights between maximum (dotted) and minimum (solid) in meters above sea level. Lines capped with square ends represent the base of each well. It is important to note the different rock type at each site, the presence of a perched aquifer in well 2D, the disconnection between site 1 and 2 wells across the same elevation, as well as the absence of water in site 3 wells.**






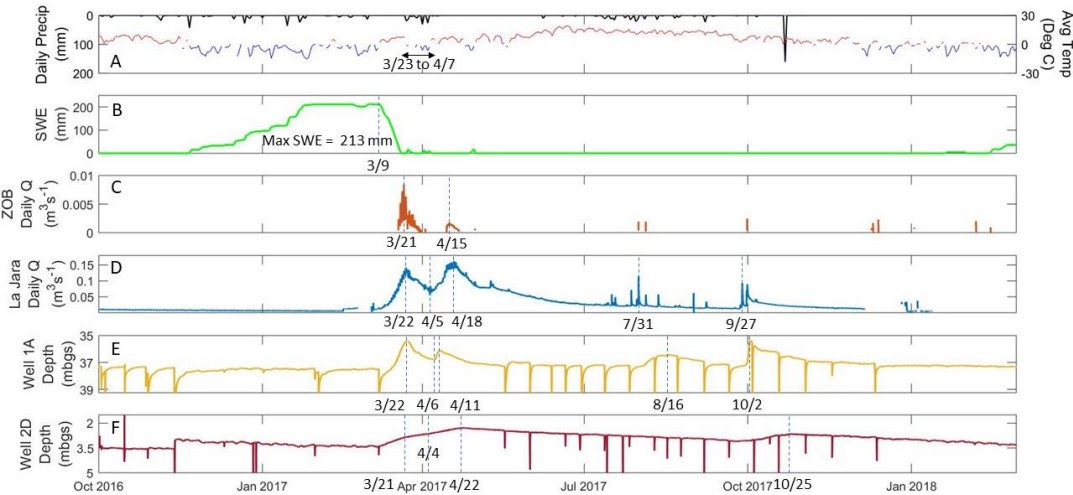


**Figure 3: A) Daily averaged precipitation falls down from top axis. Average daily temperature is shown in red when temperatures are above zero and blue at or below zero degrees Celsius. B) Snow water equivalent (SWE) for WY 2017 reaches a maximum of 213 mm. C) Fifteen-min discharge of ZOB flume for WY 2017. D) Fifteen-min discharge of La Jara flume for WY 2017. E) Well 1A depth of water below ground surface from vibrating wire piezometer placed at 41.5 m below ground surface in fifteen-min intervals. Drops in water depth correspond to sample collection. F) Well 2D depth of water below ground surface from vibrating wire piezometer placed at 6.4 m below ground surface in fifteen-min intervals. Drops in water depth correspond to sample collection.**



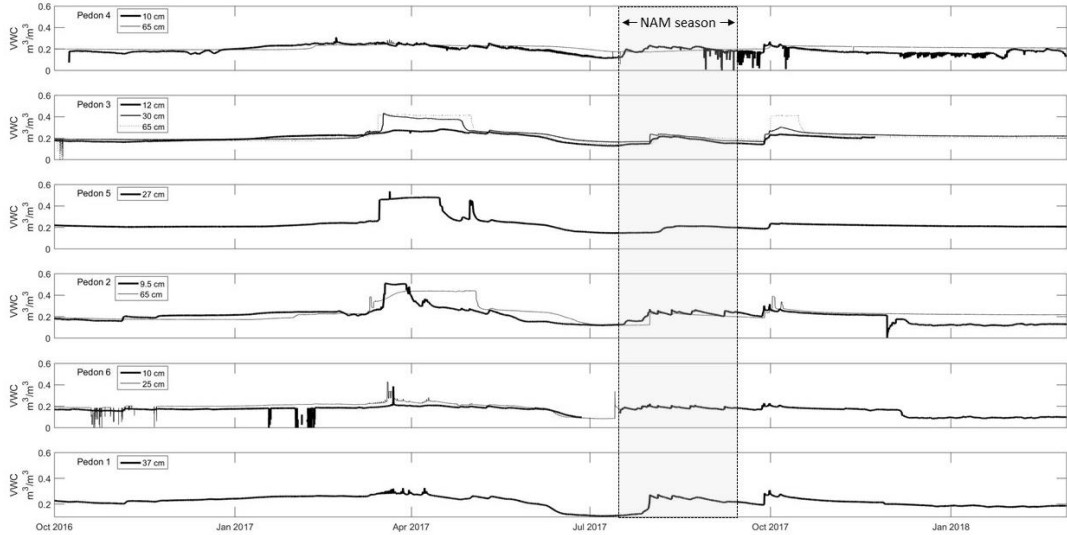

**Figure 4. Volumetric water content (VWC) of six soil pedons in the ZOB. Pedons are grouped spatially with pedons 4 and 3 on the western side of the ZOB (near site 2 wells), pedons 5 and 2 in the convergent zone (near site 3 wells), and pedons 6 and 1 on the eastern side of the ZOB (near site 1 wells). The shaded region marks the timing of North American monsoon (NAM) season from July 15th through September 15th.**






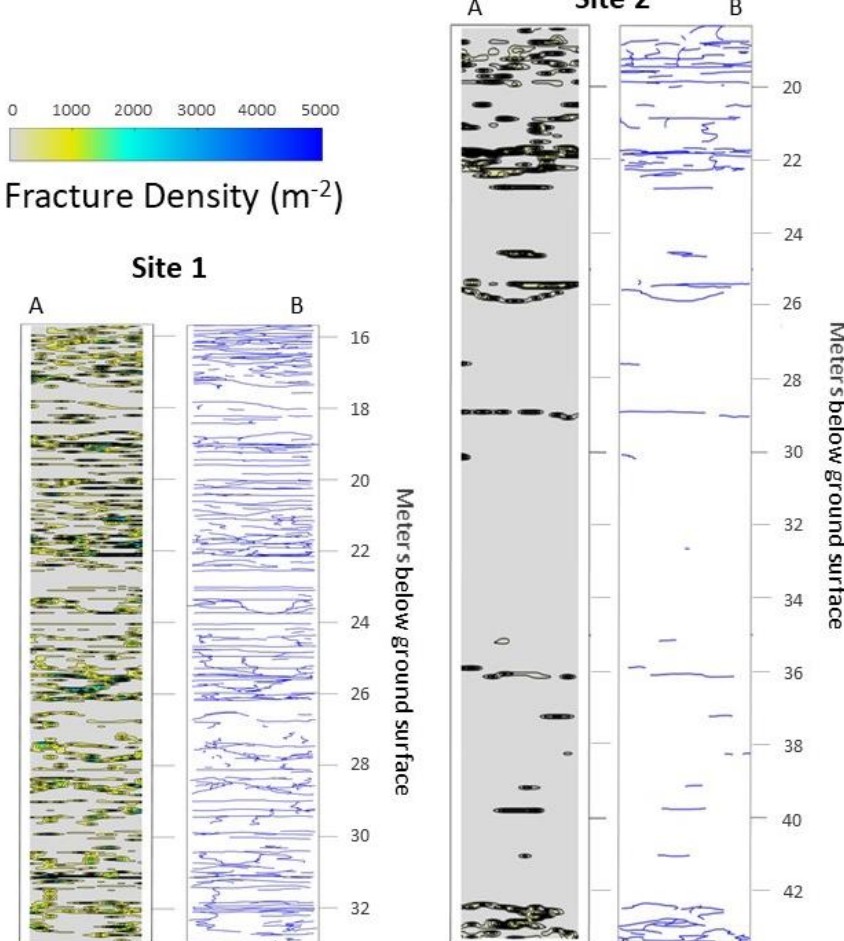

**Figure 5: A) Fracture density of sites 1 (left) and 2 (right) in m$^{-2}$. Notice that fracture density is approximately 5 times greater in site 1 than 2 according to the color scale. B) Fracture traces of sites 1 (left) and 2 (right) show where fractures exist in meters below ground surface.**






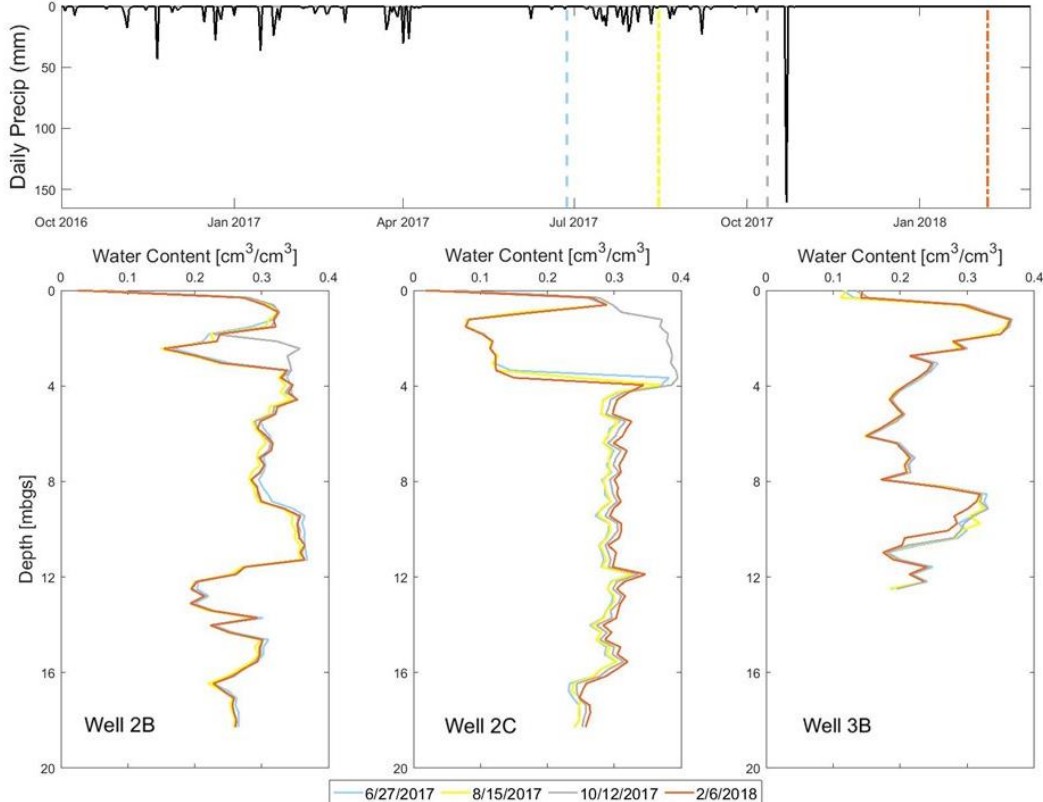

**Figure 6: Daily precipitation in mm from Redondo Weather Station above profiles of water content with depth in meters below ground surface for wells 2B, 2C, and 3B. Colors of profiles correspond to timing shown on precipitation figure above. It is important to note that the elevation above sea level of site 2 (3024 masl) and 3 (2989 masl) wells are different.**




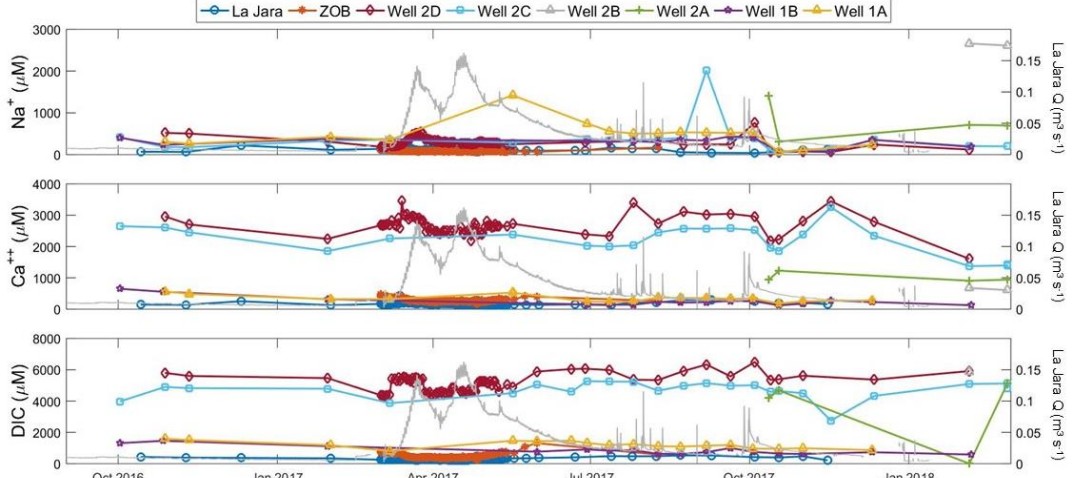

**Figure 7: Time series of Na$^+$, Ca$^{2+}$, and DIC over WY 2017 for surface waters from La Jara flume and ZOB flume and groundwaters from sites one and two. Concentrations are plotted over La Jara discharge in grey to highlight temporal changes.**


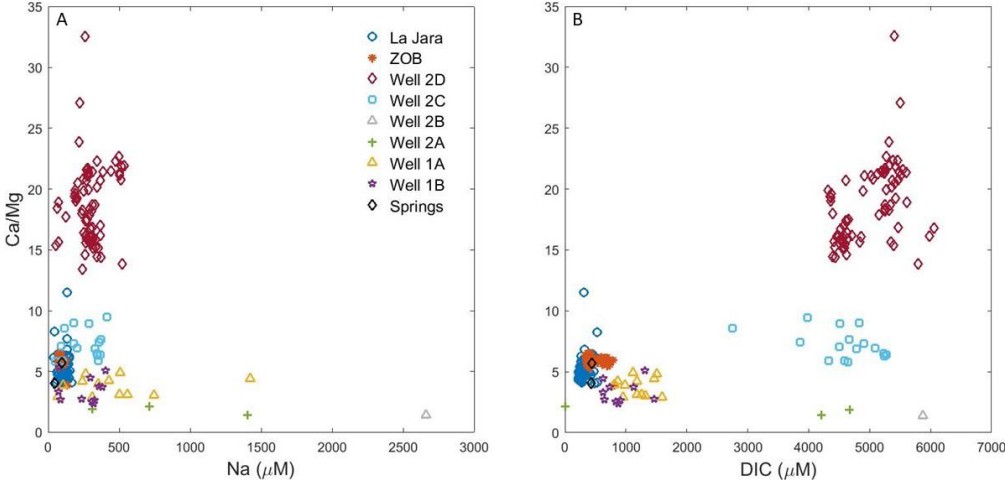

**Figure 8: A) Ca$^{2+}$/Mg$^{2+}$ molar ratios compared to Na$^+$ concentrations show that groundwaters from well 2D have Ca$^{2+}$/Mg$^{2+}$ ratios that are chemically distinct from the other water stores, which suggest that deeper groundwater is more representative of streamflow than the perched aquifer. B) Ca$^{2+}$/Mg$^{2+}$ ratios compared to**
**DIC concentrations show that site 1 groundwaters are more representative of contribution to streamflow.**





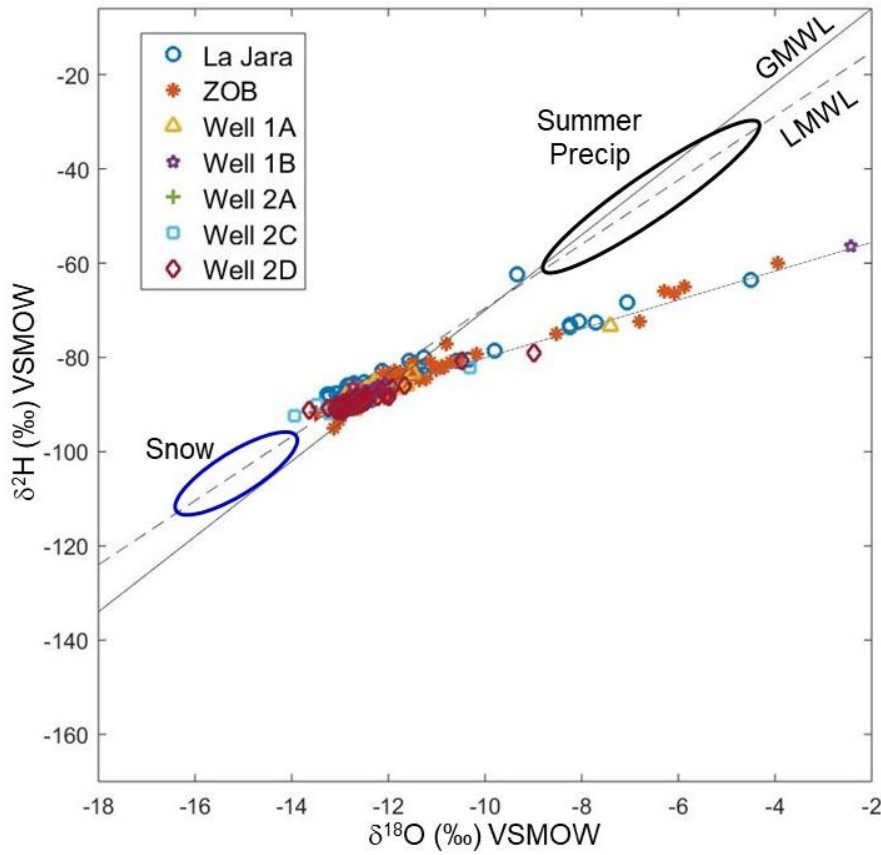

**Figure 9: Stable water isotopes ($\delta^{18}$O vs $\delta^2$H) of surface waters from La Jara and ZOB flumes, groundwaters, and volume weighted mean ranges of summer precipitation (precip) and snow. The global meteoric water line (GMWL, solid line, slope 8.0) and local meteoric water line (LMWL, slope of 6.8, dashed line from Broxton et al., 2009) are plotted for reference. Surface waters and site 1 groundwaters plot along an evaporation trend (dotted line, slope of 3.4).**






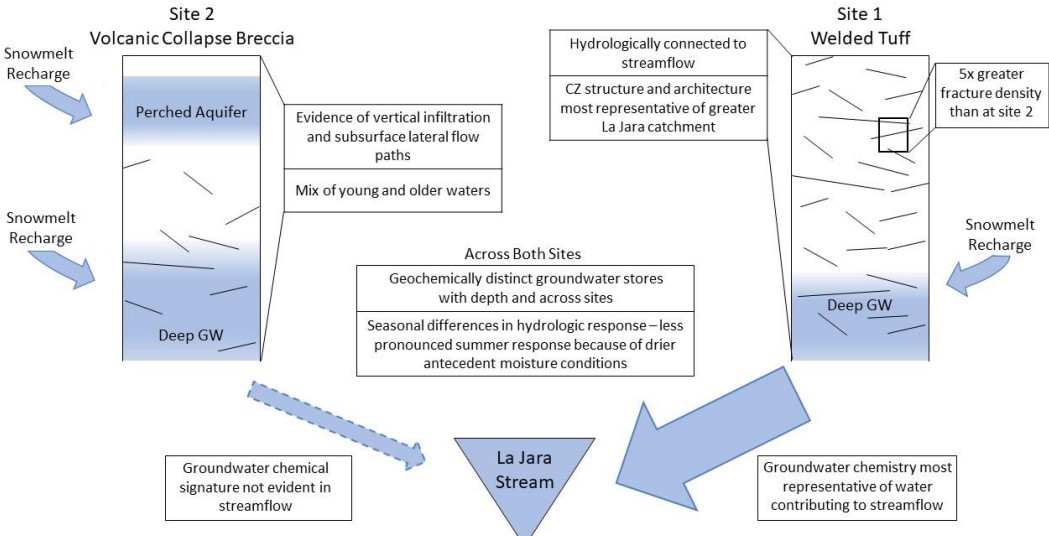

**Figure 10. Schematic of intricacies of hydrologic structure and function across two contrasting sites within the ZOB (Sites 1 and 2). Site 2 has multiple, separate stores of groundwater across depth that are distinct from each other and distinct from deep groundwater stores at site 1. All groundwater is recharged via snowmelt and seasonal differences in hydrologic response to precipitation inputs exist at both sites with less pronounced response to summer monsoons. There is evidence of vertical infiltration and subsurface lateral flow at site 2 and a mix of young and older waters, which are expected to persist across all groundwater stores. The fracture density at site 1 is approximately 5 times greater than at site 2 and the CZ structure and architecture of site 1 is most representative of the greater catchment. Deep groundwater from the fractured aquifer at site 1 is hydrologically connected to streamflow and site 1 deep groundwater chemistry is most representative of water contributing to streamflow while the distinct chemical signature of shallow groundwater from site 2 is not evident in streamflow.**








**Tables:**

| Well 2D - Volcanic Collapse Breccia | | | Well 1A - Welded Tuff | | |
|---|---|---|---|---|---|
| Date | $K_{sat}$ [m day$^{-1}$] | Mean $K_{sat}$ | Date | $K_{sat}$ [m day$^{-1}$] | Mean $K_{sat}$ |
| 6/20/2017 | $7.46 \times 10^{-3}$ | | 6/20/2017 | $1.19 \times 10^{-4}$ | |
| 6/28/2017 | $5.33 \times 10^{-3}$ | $7.22 \times 10^{-3}$ | 6/29/2017 | $1.38 \times 10^{-4}$ | $1.22 \times 10^{-4}$ |
| 7/12/2017 | $8.89 \times 10^{-3}$ | | 7/12/2017 | $1.07 \times 10^{-4}$ | |

**Table 1: Estimated saturated hydraulic conductivity in m day$^{-1}$ for three sampling events from wells 2D and 1A and their mean. Estimates were made by curve fitting 15-min VWP data with the KGS model in Aqtesolv.**

| | n | $Ca^{2+}$ (µmol/L) | | $Mg^{2+}$ (µmol/L) | | $Na^+$ (µmol/L) | | $K^+$ (µmol/L) | | DIC (µmol/L) | | pH | |
|---|---|---|---|---|---|---|---|---|---|---|---|---|---|
| La Jara Flume | 160 | 176.3 | *(39.3)* | 35.0 | *(5.6)* | 108.6 | *(20.2)* | 36.7 | *(38.5)* | 289.4 | *(58.3)* | 7.08 | *(0.37)* |
| ZOB Flume | 140 | 281.3 | *(54.9)* | 48.1 | *(11.8)* | 84.8 | *(10.1)* | 50.3 | *(13.3)* | 485.9 | *(148.0)* | 6.99 | *(0.40)* |
| Well 1A | 18 | 337.6 | *(106.0)* | 89.3 | *(32.2)* | 476.2 | *(300.0)* | 53.1 | *(9.1)* | 1198.5 | *(225.0)* | 7.55 | *(0.27)* |
| Well 1B | 16 | 254.3 | *(144.0)* | 72.7 | *(39.1)* | 286.6 | *(117.0)* | 57.1 | *(22.6)* | 851.3 | *(250.0)* | 7.21 | *(0.42)* |
| Well 2A | 4 | 1003.0 | *(130.0)* | 542.6 | *(112.0)* | 781.6 | *(394.0)* | 57.2 | *(15.1)* | 3502.8 | *(2046.0)* | 7.82 | *(0.15)* |
| Well 2B | 2 | 643.3 | *(34.1)* | 466.0 | *(15.6)* | 2636.4 | *(25.0)* | 50.2 | *(3.4)* | 5870.5 | *n = 1* | 8.21 | *(0.55)* |
| Well 2C | 24 | 2226.0 | *(453.0)* | 319.1 | *(70.5)* | 349.8 | *(381.0)* | 22.8 | *(13.4)* | 4702.5 | *(545.7)* | 7.59 | *(0.18)* |
| Well 2D | 84 | 2663.0 | *(283.8)* | 146.9 | *(22.7)* | 298.4 | *(110.4)* | 44.6 | *(12.9)* | 5083.1 | *(524.8)* | 7.51 | *(0.30)* |

**Table 2: Number of samples (n) and average concentrations of major ions and pH of surface and groundwaters.**
**Standard deviations are shown in italicized parentheses.**






| | | Tritium Analysis | | | | | | Radiocarbon Analysis | | | |
|---|---|---|---|---|---|---|---|---|---|---|---|
| | | June 2017 | | | Feb 2018 | | | Feb 2018 | | | |
| | Depth to Water (mbgs) | Tritium Content (TU) | Mean Residence Time Piston Flow Model | Mean Residence Time Exponential Model | Tritium Content (TU) | Mean Residence Time Piston Flow Model | Mean Residence Time Exponential Model | $\delta^{13}C_{DIC}$ (‰) | $^{14}C$ (pmC) | Uncorrected $^{14}C$ age ($A_0 = 100$ pmC) | Corrected $^{14}C$ age Mixing Model |
| Well 2D | 2.3 to 3.5 | 4.1 ± 0.17 | 13 ± 1.6 | 20. ± 1.8 | 4.4 ± 0.27 | 12 ± 1.8 | 17 ± 1.7 | -13.1 | 75.34 ± 0.19 | 2340 | 621 |
| Well 2C | 27.9 to 28.4 | 1.0 ± 0.19 | 39 ± 3.7 | 140 ± 28 | 0.7 ± 0.18 | 45 ± 4.8 | 202 ± 55 | -12.4 | 60.02 ± 0.17 | 4220 | 2050 |
| Well 2B | 37.6 | | | | 0.7 ± 0.20 | 45 ± 5.3 | 202 ± 60. | | | | |
| Well 2A | 45.1 to 45.7 | | | | 1.6 ± 0.21 | 30 ± 2.8 | 79 ± 12 | | | | |
| Well 1B | 35.3 to 38.0 | 2.0 ± 0.17 | 26 ± 2.1 | 59 ± 7.0 | 2.4 ± 0.24 | 23 ± 2.3 | 46 ± 6.0 | | | | |

**Table 3: Tritium content ± lab estimated error and mean residence time calculated with piston flow and exponential models ± age calculation uncertainty of groundwater from wells 2D, 2C, 2B, 2A, and 1B from two low flow sampling events (June 2017 and February 2018). Radiocarbon analysis (pmC of $^{14}C$ ± lab estimated error and $\delta^{13}C$) of dissolved inorganic carbon from wells 2D and 2C with uncorrected $^{14}C$ age calculated with $A_0$ of 100 pmC and corrected.**
