# Peer review of "Distinct stores and routing of water in the deep critical zone of a snow dominated volcanic catchment"

_Hydrology and Earth System Sciences, 2019_

## Referee Comment (RC1) · Anonymous Referee #1 · 23 May 2019

General Comments

This study employed physical measurements of stream discharge, groundwater heads, and vertical variability of water content, combined with hydrochemical and isotopic measurements to help understand the functioning of water in the critical zone at the Jemez CZO over a one-year period. The authors conclude, rather surprisingly, that a deep aquifer in fractured tuff is the principal contributor to streamflow and is better connected to recharge/discharge than shallower perched or soil-zone water reservoirs. The study has a reasonably broad base of data from which to make inferences, and in general the reasoning is clear and the final inferences seem sound. My only real

criticism is that the findings do not seem very generalizable. It is nice to know that this one, very small, catchment in the Jemez Mountains functions in this particular way, but what can we take away from this study that can be more generally applied? Does it help us, even in part, to answer larger questions that have been raised previously?

Detailed Comments

137: "uplifted" is really not the best word to describe formation of rhyolite domes; "emplaced" would be better.

140: Describing the Bandelier Tuff as "Pleistocene aged" is redundant; "Pleistocene" is a time interval and "aged" is not needed.

198: "Isotopes" are defined as variations of an element characterized by different numbers of neutrons. Water is a molecule, not an element, and therefore water does not have isotopes. "Isotopologues" is the correct terminology

245: Equations 4 and 6 do not actually give the corrected age until they are solved for "t". Equation 5 does not give the 13C fraction from carbonate dissolution; it gives the 13C fraction from atmospheric carbon.

280: What principle does the Decagon EC-5 soil-moisture sensor work on?

428: The statement "both $Ca^{2+}$ and DIC concentrations of shallow groundwater increase simultaneously, which is consistent with calcite dissolution..." is puzzling. In general, calcite will dissolve more when $Ca^{2+}$ and DIC concentrations decrease, not increase. If the system contains no calcite and it is introduced, then its dissolution will be marked by increases in concentration, but we are dealing here with a system where the calcite is presumably fixed in the rock matrix.

449: "Isotopologues" rather than "isotopes". . (Same thing for the Figure 9 caption). Also, the compositions of the isotopologues are plotted in terms of their deltaD/delta18O abundance, not "in space".

462-463: Note that the exponential ages are greater than the period over which the tritium input has remained constant (since 1992). The calculated ages are thus going to be biased young because the actual input TU was greater than the assumed.

532: The meaning of "cubic shape of the rising water table" is not clear.

556-560: The idea is not well expressed. It is the drying out between precipitation events that inhibits the infiltration of water, not "episodic recharge".

568-570: I'm not clear on the reasoning here. The much higher water content observed between 1.5 and 4.0 m during the October survey (Well 2C; Fig. 6) has to be due to infiltration of precipitation over a long period of time. The difference in volumetric water content between October and the other surveys is roughly 0.25. Over 250 cm of vadose zone, this amounts to about 600 mm of water. What is the total July-October precipitation? I doubt that it amounts to 600 mm. Certainly the 0.25 mm immediately antecedent precipitation is irrelevant! So where this water came from is something of a mystery.

583: What is the value of the "depth corresponding to the gravel-like layer"?

587: The text repeatedly refers to "lenses" of high water content. Given that these are evidenced only on 1-D vertical profiles, how can you know that they are shaped like lenses in 3-D?

600: What is meant by a "blind fault"? Usually this indicates a fault that does not outcrop at the surface, and thus would not appear on a geological map.

613: "positive Si concentration pulses" is very awkward. Why not say "found pulses of high Si concentration..." instead? The sentence is run-on and its meaning hard to decipher.

621-622: "are produced by calcite dissolution" is preferable to "are a function of calcite dissolution". Calcite as a mineral may be present in the perched aquifer, but calcite dissolution is a process that is "active" or "operative" or some other active verb. Give

the number(s) for the figure you are referring to in this paragraph.

650-653: If the Site 2 water is not found in La Jara Stream, then how does it discharge? It must leave the system somehow.

656: "isotopologues" rather than "isotopes"

657: idem

679-684: By far the most diagnostic indicator of geothermal water in the Jemez is elevated Li. Was Li measured?

688: "isotopologues"

688: I'm not sure that the extent and permeability of some of these "stores" qualifies them to be termed "aquifers".
* * *

---

## Referee Comment (RC2) · Anonymous Referee #2 · 28 Jun 2019

This manuscript presents various data to show how critical zone "structure" influences hydrologic "function" by comparing two sites with distinct lithologies and positions within the Jemez River Basin Critical Zone Observatory. Supported by geochemical / isotopic tracers and hydrologic data, the authors found that the site with highly fractured tuff had fast responses to precipitation and contributed most of the streamflow water from its deep groundwater stores, while the site with collapse breccia included disconnected perched water table aquifer that contributed little to the stream. This study presents an impressive amount of data and analysis. However, as the manuscript is currently written, it is easy for the reader to feel a bit lost about what to focus on. I provide the following suggestions for what to clarify and potentially reorganize.

1. Clarify what is referred to as critical zone "structure" and what aspect of "structure" is the focus of the study. The abstract states the main goal of the study to be to show how critical zone "structure" controls hydrologic response, but a specific definition is not provided until Research Question 2 at the end of the Introduction – it should be stated earlier. Also, the authors alternate between "structure" and "architecture" but do not explain if these refer to the same thing or not. Finally, I am guessing that "structure" and "architecture" refer to physical properties. In the definition of "structure" in Resarch Question 2, the authors include "mineralogy", but I don't see any argument for how mineral composition affects physical flow – only how it affects water chemistry, which is used as a tracer for flow.

2. Clarify what hydrologic "functions" are the focus of the study. The authors do a good job of listing functions in their abstract (water routing, storage, mean water residence times, and hydrologic response), and these correspond to some of the subsection titles of the Results and Discussions sections. However, the research questions only seem to list the two functions of "hydrologic response" and "groundwater contributions to streams", and not all of the subsection titles of the Results and Discussion correspond to the 4 functions listed in the abstract. One especially confusing aspect is that "storage" is highlighted in the manuscript title, but results mostly focus on different categories of groundwater stores, but not on any storage quantification.

3. Explain the broader implications of this work. The conclusions are very specific about what is occurring at JRB-CZO, and it would be good if the authors can comment on whether this understanding corroborates, challenges, or adds to what is already known about catchment behavior. One particular question I have is about the importance of the conclusions. Was it not to be expected that the fractured site would have faster response times? However, I do find it interesting that the perched water table aquifer is mostly disconnected to the stream – how commonly is this seen? What about the "structure" makes this disconnection happen?

4. I suggest that the authors either combine their Results and Discussions sections, or

they reorganize them so that they are more distinct. Right now, with identical subsection titles, there is much repetition in places, and the reader has to keep flipping back and forth to match up results and discussion. Also, there are a lot of laborious details in the Results section – the authors could simply point to the figures (for example, no need to point out all the specific dates and discharge values in Section 3.1).

Minor points:

P4 L104-105: I'm confused. By definition, aren't springs comprised of groundwater? If it is not groundwater, then what is the water source? Also, how is this relevant to the following Research Questions.

P5, L136: Define VCNP. Fig 1 and throughout text: I suggest naming your wells in a way so that it is easier to keep track of where they are. For example, "Well 1" could be "Well T" for Tuff, and "Well 2" could be "Well B" for Breccia.

Fig 1A: Improve resolution of text.

Fig 1B: Change the color of stream line. It is not visible with the current color and transparency.

P6, L176-177: Why were different pumping methods used at the different wells?

P6, L194: For "not shown here" - either entirely omit mention of it from the paper if it does not affect your conclusions, or put in supplementary info.

P7, L223-224: how were the uncertainties associated with the background TU concentration and measured TU in samples estimated?

P8, L260: If only showing data for 2 sites, is it necessary to mention the other ones? This also applies to P9, L307.

P9, L281: Seems like Figure 4 reference precedes Figure 3 in the text.

P9, L309: What is the "node file"?

P10, L331: Even though the water level in well 1A raises less in 2nd snowmelt event than the 1st one, and in well 2D it increases with a lower rate than the 1st snowmelt event, the discharge goes higher in La Jara stream on 4/18 relative to 3/22. Could you explain that?

Fig 3 and 4: Show the NAM time period in Figure 3 in the same way as in figure 4. Change the x-axis label to monthly intervals. Use the same scale and width for the x-axes in these two figures for easier comparison.

Fig4 , pedon 3: VWC at 65cm depth is hard to see.

P11, L361: Explain why changes in VWC are more pronounced in deeper parts. Why is the response for pedon 5 different than for pedons 1 and 6, even though they seem to have the same geology based on their locations on fig 1?

P13, L420: Could you explain why major ion concentrations are so different in wells 2A and 2B relative to 2C and 2D? If 2D is a perched aquifer with vertical connection to the wells beneath as the author mentioned in P8, L629, the temporal changes in the major ion concentrations should follow the same trend, but that is not seen in the figure.

Fig 7: It is difficult to see the trend with lines with markers. Removing markers could make it easier to read.

P13, L433: Briefly explain why Na+ concentration increases in well 1A around June 2017 (again it would be helpful if the x-axis labels are at monthly intervals).

P14, L454: Provide discussion about the enrichment.

P14, L479-482: This sentence should be reworked. It currently implies that understanding the geochemistry is the end-goal, but actually, the geochemistry is the means for understanding the impacts of the structure and architecture. The logic in the current wording seems backwards.

P14, L497 paragraph: Is there a way to back out K values that are more relevant for

the spatial scale of interest, which should be higher in tuff than breccia? For example, using the discharge rates and hydraulic gradients? Would the backed out K values be more consistent with literature values for high vs. less dense fractures than the slug test K results?

P15, L518: Typo: sentence ends with "and"

P16, L539: Maybe "in contrast to" instead of "however"?

P16, L546: Isn't lesser water table response to summer rains typically due to higher ET, which prevents wetting fronts to descend below the root zone?

P17, L569: delete comma after "both"

P17, L583: Seems like the correspondence of gravel and wetting patterns is major part of the paper's findings about the relationship between structure and hydrologic function. As such, the gravel data should be presented more prominently. At the very least, state what depth corresponds to the gravel-like layer. I would suggest to even further show graphically where the gravel is – either superimposed on Figure 6 or on a separate dedicated figure with similar y-axis scale.

P19, L642 paragraph: Seems out of sequence. Shouldn't this summary conceptual model come AFTER the subsequent section and old and young water?

Figure 9: The ellipses for the Summer Precip and Snow seem very approximate. Is there a more specific range?

P19, L665: Figure 10 is referenced, but it seems like a figure showing tritium results should be referenced instead. Is there supposed to be a figure showing tritium measurements?

P20, L703-704: parenthetical for "structure" includes "deep groundwater" and "longer mean residence time", but neither of those are properties of the physical porous media. I assumed "structure" to refer to to the physical porous media.

---

## Author Comment (AC1) · 5 Aug 2019

Storage and routing of water in the deep critical zone of a snow dominated volcanic catchment by Alissa White et al. Authors' reply to Anonymous Referee #1

General Comments This study employed physical measurements of stream discharge, groundwater heads, and vertical variability of water content, combined with hydrochemical and isotopic measurements to help understand the functioning of water in the critical zone at the Jemez CZO over a one-year period. The authors conclude, rather surprisingly, that a deep aquifer in fractured tuff is the principal contributor to streamflow and is better connected to recharge/discharge than shallower perched or soil-zone

water reservoirs. The study has a reasonably broad base of data from which to make inferences, and in general the reasoning is clear and the final inferences seem sound. My only real criticism is that the findings do not seem very generalizable. It is nice to know that this one, very small, catchment in the Jemez Mountains functions in this particular way, but what can we take away from this study that can be more generally applied? Does it help us, even in part, to answer larger questions that have been raised previously?

We appreciate the referee's thoughtful comments and helpful suggestions. We have addressed their specific comments by clarifying text relating to volumetric water content and oxygen and hydrogen isotopes, as well as, accepting their suggestions for more accurate word choice. We have also addressed their broader comment by adding paragraphs to our concluding remarks that stresses the implications, general utility, and potential application of our findings to other studies in mountain systems. Detailed responses to each of the referee's comments are below each comment.

Added to conclusions to stress implications: Surprisingly, deep groundwater from site 1 wells appear to be more chemically-representative of waters that contribute to La Jara stream and more representative of the structure (geology, fractured aquifer, and greater depth to water table) and function (hydrologic response, solute fluxes, and water routing) of the CZ in the greater La Jara catchment, suggesting that deep groundwater from the fractured aquifer, rather than shallow subsurface flow from the perched aquifer, sustains stream baseflow. Further, we suggest that the deep subsurface flow paths observed in the JRB-CZO are likely a signature of snow dominated volcanic catchments transferable to other deeply fractured extrusive bedrock systems. The dominant contribution of deep groundwater to surface flows and the hydraulic connection between the fractured bedrock aquifer and streamflow may suggest that deep groundwater stores in fractured bedrock aquifers are sensitive to changes in climatic drivers of streamflow like shifts in precipitation magnitude and timing, as predicted in the southwestern United States. We assert that this study emphasizes the utility of interdisciplinary research

to discern the distribution of groundwater stores, their connection to streamflow, and the underlying impact of CZ architecture on hydrologic response to climatic drivers. Furthermore, we propose that it highlights the need to better characterize the deep subsurface of mountain systems by transferring this approach to other complex settings that challenge and advance the current understanding of subsurface hydrologic systems around the world. We hope that this study provides an example of how to bring together multiple lines of evidence to simultaneously examine both, CZ architecture and hydrology, through hydrometric, geophysical, geochemical, and residence time analyses.

Detailed Comments 137: "uplifted" is really not the best word to describe formation of rhyolite domes; "emplaced" would be better. "Uplift of" changed to "emplacement of"

140: Describing the Bandelier Tuff as "Pleistocene aged" is redundant; "Pleistocene" is a time interval and "aged" is not needed. We agree and "aged" has been removed.

198: "Isotopes" are defined as variations of an element characterized by different numbers of neutrons. Water is a molecule, not an element, and therefore water does not have isotopes. "Isotopologues" is the correct terminology Rather than using isotopologues of water, this text uses isotopes of both elements of water (oxygen and hydrogen stable isotopes) measured independently of one another; therefore, we do not believe that the suggested change to isotopologues should be made. However, we have clarified the use of "stable water isotopes" throughout the text to refer specifically to "d18O and d2H" instead.

245: Equations 4 and 6 do not actually give the corrected age until they are solved for "t". Equation 5 does not give the 13C fraction from carbonate dissolution; it gives the 13C fraction from atmospheric carbon. Equations 4 and 5 were solved for "t" and the title of Equation 5 was changed.

280: What principle does the Decagon EC-5 soil-moisture sensor work on? A few lines were added to the methods section 2.6 to explain the scientific principle underlying the

function of Decagon EC-5 sensors.

428: The statement "both Ca2+ and DIC concentrations of shallow groundwater increase simultaneously, which is consistent with calcite dissolution. . ." is puzzling. In general, calcite will dissolve more when Ca2+ and DIC concentrations decrease, not increase. If the system contains no calcite and it is introduced, then its dissolution will be marked by increases in concentration, but we are dealing here with a system where the calcite is presumably fixed in the rock matrix. We reworded this sentence to clarify. We find that both Ca2+ and DIC concentrations increase and suggest that those increases in concentration are the result of calcite dissolution. We posit that Ca2+ and DIC concentrations increase as calcite dissolves around the onset of spring snowmelt and return to lower concentrations as groundwater becomes saturated with respect to calcite causing Ca2+ and DIC to precipitate out of solution. We make this suggestion based on calcite saturation index calculations of shallow groundwater from previous work (Olshansky et al., 2018) at the JRB-CZO. Time series of the calcite saturation index of shallow groundwater display a marked increase from near equilibrium at the onset of spring snowmelt (coincident with the time that the current study notes increased Ca2+ and DIC concentrations) to saturated around April 1 (when the current study notes the onset of decreasing Ca2+ and DIC concentrations).

449: "Isotopologues" rather than "isotopes". . (Same thing for the Figure 9 caption). Also, the compositions of the isotopologues are plotted in terms of their deltaD/delta18O abundance, not "in space". See previous comment about the rewording of the phrase "stable water isotopes" throughout the text. The description of isotope values in space was corrected.

462-463: Note that the exponential ages are greater than the period over which the tritium input has remained constant (since 1992). The calculated ages are thus going to be biased young because the actual input TU was greater than the assumed. We agree that residence times calculated from tritium concentrations are biased towards shorter times and have added this note into the methods section 2.3 to accompany the

description of residence time calculations.

532: The meaning of "cubic shape of the rising water table" is not clear. We more clearly direct the reader to Figure 3F to see the cubic shape of the well 2D hydrograph during snowmelt 2017 and describe the shape of the curve. This cubic shape is characterized by a gradual increase in groundwater level followed by an inflection point and subsequent more rapid rise in groundwater level as described by Sophocleous et al., 1988, 1991a, 1991b.

556-560: The idea is not well expressed. It is the drying out between precipitation events that inhibits the infiltration of water, not "episodic recharge". We acknowledge the point that the reviewer is making here, but we argue that small and frequent rain events facilitate and exacerbate more frequent drying out events, which mirrors the "episodic recharge" scenario of Langston et al. (2015). Langston et al. (2015) found that less recharge reached the groundwater table when precipitation was spread over several, small events each followed by a period of evaporation as compared to one concentrated event that mimics seasonal melt and subsequent evaporation. They concluded that the timing of recharge exerted a strong control on the degree of saturation and flow paths within the subsurface and that frequent wetting and drying inhibits fluid flow deep into the subsurface. These modeled scenarios strongly resemble the differences our study sees in soil moisture content during NAM season and spring snowmelt.

568-570: I'm not clear on the reasoning here. The much higher water content observed between 1.5 and 4.0 m during the October survey (Well 2C; Fig. 6) has to be due to infiltration of precipitation over a long period of time. The difference in volumetric water content between October and the other surveys is roughly 0.25. Over 250 cm of vadose zone, this amounts to about 600 mm of water. What is the total July-October precipitation? I doubt that it amounts to 600 mm. Certainly the 0.25 mm immediately antecedent precipitation is irrelevant! So where this water came from is something of a mystery. We acknowledge that the 0.25 cm3 cm-3 change in VWC is large compared to the precipitation inputs immediately prior to the October NP measurement. We have

changed the precipitation inputs immediately prior to each NP event to cumulative precipitation since the last NP event. For instance, the cumulative precipitation between the June and August NP measurements is 168.13 mm while the cumulative precipitation between August and October NP measurements is 67.55 mm, making the total precipitation input between June and October NP measurements 235.68 mm. The 600 mm mentioned by the referee is very near the cumulative annual precipitation for WY 2017 of 637.5 mm and close to the average annual precipitation from 1981-2012 of 711 mm (Zapata-Rios et al., 2015b). Surprisingly, cumulative precipitation between June and August is greater than cumulative precipitation between August and October. Again, we return to our hypothesis that frequent wetting and drying inhibits infiltration into the subsurface. It is possible that the Redondo Weather Station, while only 1.5 km from the wells in the ZOB, does not capture representative precipitation inputs because of elevation differences (Redondo Weather Station at 3231 masl while site 2 wells at 3024 masl) and the sporadic nature of NAM storms. It is also possible that lateral subsurface flow contributes some quantity of water to well 2C between the August and October NP measurements; however, we think this is unlikely. The shape of the VWC curve gradually increasing from ground surface down to its max at 4 mbgs, the generally lower VWC (in 3 of 4 NP measurements) at the surface compared to the generally higher VWC below 4 mbgs not indicative of lateral flow, the increased shallow soil water content (Figure 4) during the time of the October NP measurement, and the fact that the shallow groundwater water table is rising during the time of the October NP measurement suggests vertical infiltration rather than subsurface lateral flow.

583: What is the value of the "depth corresponding to the gravel-like layer"? The depth of 1.5 to 2.3 mbgs was added to the text and Figure 6.

587: The text repeatedly refers to "lenses" of high water content. Given that these are evidenced only on 1-D vertical profiles, how can you know that they are shaped like lenses in 3-D? We did not mean to imply a 3D shape of the "lenses" and have changed to words like layer and zone that do not imply 3D shape.

[Figure]

600: What is meant by a "blind fault"? Usually this indicates a fault that does not outcrop at the surface, and thus would not appear on a geological map. We have changed the use of "blind fault" to "concealed fault" in the text. The authors used the term blind fault to refer a fault that is noted as certain and concealed in the geologic map of Goff et al. (2011). We have used the geologic map of Goff et al. (2011) in the analysis and figures herein; however, Goff et al. (2011) notes that they did not substantially change the primary fault patterns of previously maps from Smith and Bailey (1968) and Smith et al. (1970).

613: "positive Si concentration pulses" is very awkward. Why not say "found pulses of high Si concentration. . ." instead? The sentence is run-on and its meaning hard to decipher. This change was made and the sentence was reworded.

621-622: "are produced by calcite dissolution" is preferable to "are a function of calcite dissolution". Calcite as a mineral may be present in the perched aquifer, but calcite dissolution is a process that is "active" or "operative" or some other active verb. Give the number(s) for the figure you are referring to in this paragraph. Changes were made.

650-653: If the Site 2 water is not found in La Jara Stream, then how does it discharge? It must leave the system somehow. We are unable to conclude where the site 2 water drains. Text detailing the possibility of the perched groundwater transmitting to streamflow was added (Lines 668-670).

656: "isotopologues" rather than "isotopes" See previous comment about the rewording of the phrase "stable water isotopes" throughout the text.

657: idem See previous comment about the rewording of the phrase "stable water isotopes" throughout the text.

679-684: By far the most diagnostic indicator of geothermal water in the Jemez is elevated Li. Was Li measured? Unfortunately, Li concentrations were not measured.

688: "isotopologues" See previous comment about the rewording of the phrase "stable

water isotopes" throughout the text.

688: I'm not sure that the extent and permeability of some of these "stores" qualifies them to be termed "aquifers". We no longer refer to the site 2 deeper groundwaters as aquifers and reserve the use of that term for the perched aquifer (well 2D) and site 1 deep fractured aquifer.

---

## Author Comment (AC2) · 5 Aug 2019

Distinct stores and routing of water in the deep critical zone of a snow dominated volcanic catchment by Alissa White et al. Authors' reply to Anonymous Referee #2

General Comments

This manuscript presents various data to show how critical zone "structure" influences hydrologic "function" by comparing two sites with distinct lithologies and positions within the Jemez River Basin Critical Zone Observatory. Supported by geochemical / isotopic tracers and hydrologic data, the authors found that the site with highly fractured tuff had

fast responses to precipitation and contributed most of the streamflow water from its deep groundwater stores, while the site with collapse breccia included disconnected perched water table aquifer that contributed little to the stream. This study presents an impressive amount of data and analysis. However, as the manuscript is currently written, it is easy for the reader to feel a bit lost about what to focus on. I provide the following suggestions for what to clarify and potentially reorganize.

We appreciate the referee's thoughtful comments and helpful suggestions. We have addressed their specific comments by clarifying text and addressing specific questions they brought up. We have also addressed their broader comments by explicitly defining structure and function, reorganizing our results and discussion, and adding a paragraph to our concluding remarks that stresses the implications, general utility, and potential application of our findings to other studies in mountain systems. Detailed responses to each of the referee's comments are provided below each comment.

Major Comments

1. Clarify what is referred to as critical zone "structure" and what aspect of "structure" is the focus of the study. The abstract states the main goal of the study to be to show how critical zone "structure" controls hydrologic response, but a specific definition is not provided until Research Question 2 at the end of the Introduction – it should be stated earlier. Also, the authors alternate between "structure" and "architecture" but do not explain if these refer to the same thing or not. Finally, I am guessing that "structure" and "architecture" refer to physical properties. In the definition of "structure" in Research Question 2, the authors include "mineralogy", but I don't see any argument for how mineral composition affects physical flow – only how it affects water chemistry, which is used as a tracer for flow.

A definition of CZ structure, as well as clarification of the use of structure and architecture interchangeably was added to the introduction in lines 64-66. We agree with the reviewer's point about the use of mineralogy and have removed mineralogy as one of

the examples listed in Research Question 2. Herein, we use the term subsurface structure or architecture to refer to physical properties of the subsurface such as lithology, fracture density, and location and extent of geologic heterogeneities that may impact movement of water through the subsurface.

2. Clarify what hydrologic "functions" are the focus of the study. The authors do a good job of listing functions in their abstract (water routing, storage, mean water residence times, and hydrologic response), and these correspond to some of the subsection titles of the Results and Discussions sections. However, the research questions only seem to list the two functions of "hydrologic response" and "groundwater contributions to streams", and not all of the subsection titles of the Results and Discussion correspond to the 4 functions listed in the abstract. One especially confusing aspect is that "storage" is highlighted in the manuscript title, but results mostly focus on different categories of groundwater stores, but not on any storage quantification.

The paper's title was modified from "Storage and routing of water in the deep critical zone of a snow dominated volcanic catchment" to "Distinct stores and routing of water in the deep critical zone of a snow dominated volcanic catchment" because without more extensive geophysics or drilling we are unable to estimate the lateral extent of the aquifers and; therefore, cannot quantify storage.

Research Question 2 was modified to include water routing, mean residence times, and seasonal contribution of distinct groundwater stores to streams. The hydrologic functions outlined in the abstract, research questions, and discussion section titles now match and help to clarify which hydrologic functions are investigated in this paper.

RQ2: How does CZ architecture, such as fracture density, lithology, and subsurface heterogeneities, influence water routing, mean residence times, and the seasonal contribution of distinct groundwater stores to streams?

3. Explain the broader implications of this work. The conclusions are very specific about what is occurring at JRB-CZO, and it would be good if the authors can comment

on whether this understanding corroborates, challenges, or adds to what is already known about catchment behavior. One particular question I have is about the importance of the conclusions. Was it not to be expected that the fractured site would have faster response times? However, I do find it interesting that the perched water table aquifer is mostly disconnected to the stream – how commonly is this seen? What about the "structure" makes this disconnection happen?

The unexpected finding is that deep groundwater from the fractured site was connected to streamflow year round while the perched aquifer does not contribute significantly to streamflow, although the perched aquifer was expected to be connected to the stream, and this had previously been suggested to be the case. Transmission of perched aquifers to streams has been previously documented, as described in the introduction and conceptual models typically generalize to show contributions to streams from multiple sources and depths of groundwater. The differences in lithology (including presence of confining layer) and subsequent differences in fracture density between tuff (site 1 wells) and collapse breccia deposits (site 2 wells, specifically perched aquifer) that are scattered throughout the JRB-CZO controls the connection or lack of connection between groundwater stores and streamflow.

Added to conclusions to expand implications: Surprisingly, deep groundwater from site 1 wells appear to be more chemically-representative of waters that contribute to La Jara stream and more representative of the structure (geology, fractured aquifer, and greater depth to water table) and function (hydrologic response, solute fluxes, and water routing) of the CZ in the greater La Jara catchment, suggesting that deep groundwater from the fractured aquifer, rather than shallow subsurface flow from the perched aquifer, sustains stream baseflow. Further, we suggest that the deep subsurface flow paths observed in the JRB-CZO are likely a signature of snow dominated volcanic catchments transferable to other deeply fractured extrusive bedrock systems. The dominant contribution of deep groundwater to surface flows and the hydraulic connection between the fractured bedrock aquifer and streamflow may suggest that deep groundwater stores in

fractured bedrock aquifers are sensitive to changes in climatic drivers of streamflow like shifts in precipitation magnitude and timing, as predicted in the southwestern United States. We assert that this study emphasizes the utility of interdisciplinary research to discern the distribution of groundwater stores, their connection to streamflow, and the underlying impact of CZ architecture on hydrologic response to climatic drivers. Furthermore, we propose that it highlights the need to better characterize the deep subsurface of mountain systems by transferring this approach to other complex settings that challenge and advance the current understanding of subsurface hydrologic systems around the world. We hope that this study provides an example of how to bring together multiple lines of evidence to simultaneously examine both, CZ architecture and hydrology, through hydrometric, geophysical, geochemical, and residence time analyses.

4. I suggest that the authors either combine their Results and Discussions sections, or they reorganize them so that they are more distinct. Right now, with identical subsection titles, there is much repetition in places, and the reader has to keep flipping back and forth to match up results and discussion. Also, there are a lot of laborious details in the Results section – the authors could simply point to the figures (for example, no need to point out all the specific dates and discharge values in Section 3.1).

We followed the reviewer's suggestion to reorganize the results and discussion sections to make them more distinct rather than combine them entirely. The results section titles have been changed to focus more on the type of analysis performed rather than the finding as the discussion section titles do. Many of the laborious specifics from the results section were pulled from the text. Instead, the authors point the reader to the figures and tables where those details can be found.

Minor comments

P4 L104-105: I'm confused. By definition, aren't springs comprised of groundwater? If it is not groundwater, then what is the water source? Also, how is this relevant to the

following Research Questions.

While it is true that springs are comprised of groundwater by definition (i.e., saturated zone intersection with the soil surface), the composition of that water source may be quite distinct from the composition of the groundwater sampled, e.g., as from groundwater monitoring wells, because by transport through the near surface soil media, the groundwater composition may be altered relative to its composition at depth. Therefore, Frisbee et al., (2013) found that springs integrated multiple water sources like groundwater, soil water, unsaturated flow, and precipitation. These details of integrated water sources in springs were added to the text. This point about springs is meant to highlight how the analysis of groundwater in this paper enhances previous research that used springs as proxies for groundwater at this intensively studied observatory.

P5, L136: Define VCNP. Fig 1 and throughout text: I suggest naming your wells in a way so that it is easier to keep track of where they are. For example, "Well 1" could be "Well T" for Tuff, and "Well 2" could be "Well B" for Breccia. We defined the abbreviation VCNP as Valles Caldera National Preserve. We intended the distinctions in rock type highlighted in Figures 1 and 2 to help readers keep track of the locations of each well.

Fig 1A: Improve resolution of text. Text resolution was fixed.

Fig 1B: Change the color of stream line. It is not visible with the current color and transparency. Stream line was changed to darker blue.

P6, L176-177: Why were different pumping methods used at the different wells? Text was added to clarify the different diameter casing of wells 1 and 2 and readers were directed to Supplemental Table 4.

P6, L194: For "not shown here" - either entirely omit mention of it from the paper if it does not affect your conclusions, or put in supplementary info. Those items were pulled.

P7, L223-224: how were the uncertainties associated with the background TU concentration and measured TU in samples estimated? These details were added to the methods section.

P8, L260: If only showing data for 2 sites, is it necessary to mention the other ones? This also applies to P9, L307. The first sentence (Line 260) was moved from the main text to the figure caption of Supplemental Figure 1 and Supplemental Figure 1 was referenced in the previous mention of vibrating wire piezometers. We maintain that it is important to explain the monitoring well and transducer details of all wells as this is the first paper to describe their construction and installation process so we want to keep those details in supplemental. The second sentence (Line 307) was removed from the text entirely.

P9, L281: Seems like Figure 4 reference precedes Figure 3 in the text. That reference was premature in the description of methods; therefore, the reference was changed to direct readers to the location of the pedons (Figure 1) rather than to the results from them.

P9, L309: What is the "node file"? The detailed reference to a node file (file that saves locations of fractures from Adobe Illustrator) was pulled and reworded to clarify.

P10, L331: Even though the water level in well 1A raises less in 2nd snowmelt event than the 1st one, and in well 2D it increases with a lower rate than the 1st snowmelt event, the discharge goes higher in La Jara stream on 4/18 relative to 3/22. Could you explain that?

The second snowmelt peak in La Jara streamflow exceeded the first snowmelt peak because a considerable depth of snow (approximately 500 mm; Olshansky et al., 2018) remained on the ground in La Jara catchment when temperatures dropped below freezing again in March, adding to La Jara streamflow that remained much greater than baseflow between peaks. This explanation was added to the text.

Fig 3 and 4: Show the NAM time period in Figure 3 in the same way as in figure

4. Change the x-axis label to monthly intervals. Use the same scale and width for the x- axes in these two figures for easier comparison. We added the shaded NAM time period to Figure 3 and adjusted the figures' scales for easier comparison between Figures 3 and 4. However, we think that changing labels from every 3 months to every month would make figures too crowded, especially in Figure 3 where specific dates are highlighted on the figure.

Fig4 , pedon 3: VWC at 65cm depth is hard to see. This adjustment was made.

P11, L361: Explain why changes in VWC are more pronounced in deeper parts. Why is the response for pedon 5 different than for pedons 1 and 6, even though they seem to have the same geology based on their locations on fig 1? Pedon 5, while located in the same tuff rock type as pedons 1 and 6, is situated in the convergent zone of the ZOB and therefore receives lateral subsidies of moisture from upslope, and retains more moisture than pedons 1 and 6 located upslope of the convergent zone. Text highlighting this distinction was added to the text. Changes in VWC are likely most pronounced in deeper parts of the soil pedons because of subsurface lateral flow. Neutron probe profiles show evidence of subsurface lateral flow at approximately 12 mbgs at site 2, but were unable to confidently capture those changes less than 1 mbgs because of escape of neutrons through the surface. Previous work (Liu et al., 2008; McIntosh et al., 2017; Olshansky et al., 2018) also found minimal overland flow and suggested the presence of subsurface lateral flowpaths in the JRB-CZO. This text was added to section 3.1.

P13, L420: Could you explain why major ion concentrations are so different in wells 2A and 2B relative to 2C and 2D? If 2D is a perched aquifer with vertical connection to the wells beneath as the author mentioned in P8, L629, the temporal changes in the major ion concentrations should follow the same trend, but that is not seen in the figure. Differences in mineralogy, specifically the presence of calcite within the top 15 mbgs at site 2 wells, create differences in major ion concentrations – generally higher Ca2+ and DIC in wells 2D and 2C and higher Mg2+, Na+, and K+ in wells 2A and 2B where

there are greater percentages of feldspars at depth. Unfortunately, there are far fewer samples from the deeper site 2 wells (2A and 2B) and the geochemical time series of those wells begins in October 2017 because of issues developing those deepest wells. These differences in sampling frequency prevent the comparison of temporal changes across depth in site 2 wells. We do, however, see that the concentration of major ions in wells 2D and 2C change simultaneously.

Fig 7: It is difficult to see the trend with lines with markers. Removing markers could make it easier to read. We think that the markers will be necessary to distinguish between sites for readers that print in black and white. We also think that the markers are helpful to highlight the differences in sampling frequency across time and between sites. We tried adjusting the size and shape of markers, but ultimately felt that the markers are best as they are.

P13, L433: Briefly explain why Na+ concentration increases in well 1A around June 2017 (again it would be helpful if the x-axis labels are at monthly intervals). Increased Na+ concentrations are likely the result of the weathering of feldspars, like albite, which were observed in quantitative mineralogic analysis of cores (Moravec et al., 2018). We hypothesize that site 1 Na+ concentrations increase following spring snowmelt on 5/27 as the well 1A water table slowly recedes allowing greater contact time for weathering of feldspars at approximately 37 mbgs. This explanation was added to the text.

P14, L454: Provide discussion about the enrichment. Discussion of the enrichment of ïА̌ď18O and ïА̌ď2H is provided in the discussion in lines 674 to 684.

P14, L479-482: This sentence should be reworked. It currently implies that understanding the geochemistry is the end-goal, but actually, the geochemistry is the means for understanding the impacts of the structure and architecture. The logic in the current wording seems backwards. We agree and this sentence was reworked.

P14, L497 paragraph: Is there a way to back out K values that are more relevant for the spatial scale of interest, which should be higher in tuff than breccia? For example,

using the discharge rates and hydraulic gradients? Would the backed out K values be more consistent with literature values for high vs. less dense fractures than the slug test K results? While we appreciate the suggestion and agree that it would be great to have effective K values for a larger scale, we do not think this calculation can be made with the current dataset. Because it is unclear where the perched aquifer drains and there is only one monitoring well accessing this aquifer, it is not possible to measure the hydraulic gradient or discharge needed for calculation of K in the breccia. We would need more geophysical data to discern the spatial extent of both aquifers and question the validity of the assumption that all La Jara creek discharge would be attributed solely to groundwater contribution from the fractured tuff aquifer.

P15, L518: Typo: sentence ends with "and" Typo corrected.

P16, L539: Maybe "in contrast to" instead of "however"? Change made.

P16, L546: Isn't lesser water table response to summer rains typically due to higher ET, which prevents wetting fronts to descend below the root zone? We agree that lesser water table response to summer rains is a function of increased ET, which works to produce drier antecedent soil moisture like that referred to in Figure 4 and highlighted in the text. Text was added to clarify this point.

P17, L569: delete comma after "both" Change made.

P17, L583: Seems like the correspondence of gravel and wetting patterns is major part of the paper's findings about the relationship between structure and hydrologic function. As such, the gravel data should be presented more prominently. At the very least, state what depth corresponds to the gravel-like layer. I would suggest to even further show graphically where the gravel is – either superimposed on Figure 6 or on a separate dedicated figure with similar y-axis scale. The depth of the gravel-like layer was added to the text and the layer is now highlighted in Figure 6.

P19, L642 paragraph: Seems out of sequence. Shouldn't this summary conceptual

model come AFTER the subsequent section and old and young water? We agree and that paragraph has been moved.

Figure 9: The ellipses for the Summer Precip and Snow seem very approximate. Is there a more specific range? These ranges are volume weighted means from previous studies in the JRB-CZO. The range of d18O and d2H for snow were taken directly from Gustafson, 2008 and those of summer precipitation were taken from Zapata Rios et al. (2015b). The ellipses were fit directly over that data. These specific details were more added to the methods section 2.3 and to Figure 9 caption.

P19, L665: Figure 10 is referenced, but it seems like a figure showing tritium results should be referenced instead. Is there supposed to be a figure showing tritium measurements? A reference to Table 3, which provides details of the tritium measurements was added.

P20, L703-704: parenthetical for "structure" includes "deep groundwater" and "longer mean residence time", but neither of those are properties of the physical porous media. I assumed "structure" to refer to the physical porous media. "Longer mean residence time" was removed and "deep groundwater" was changed to greater depth to water table.

---

## Author Response (AR1)

**Storage and routing of water in the deep critical zone of a snow dominated volcanic catchment by Alissa White et al.**

**Author's Reply to Editor**

Editor's Comments:

Thank you for taking the opportunity of improving your manuscript by being responsive to the reviewers' comments. I think this is shaping up into a terrific contribution, and I have a few suggestions. (1) In terms of the general contribution of this work, let me suggest another argument for its importance. In integrated hydrologic models that include mountains, the assumption of a perfunctory, shallow groundwater system only is quite common. When the suggestion is made to include deeper groundwater processes, the answer is always: K decreases with depth so why bother, or there is no data. Given this sorry state of affairs, any study that elucidates the deeper groundwater processes in mountain systems is an important contribution. (2) Instead of the term aquifer, you might consider just using the term "aquifer system." (3) I am not convinced that your "perched aquifer" is perched. From Fig. 2 it looks like dh/dz between the shallow water table and the next deeper groundwater level is about 1, which would indicate saturated vertical downward flow from the part tapped by 2D to part tapped by 2C. For it to be clearly perched, there would have to be a dh/dz > 1m or actual evidence that the intervening rocks are dry even where there is adequate porosity.

Reply to Editor's Comments

Thank you for your guidance. We are grateful for the opportunity to improve the manuscript and appreciate your recommendations. Please find our response to each comment below. The text added directly to the manuscript is highlighted in green.

1) We agree that the finding that groundwater from deep fractured aquifer systems sustains perennial mountain streamflow will have important implications for hydrologic models, especially in mountain systems where little is known about subsurface properties. Furthermore, we hope that coupling the analysis of subsurface structure and hydrologic processes in this remote high elevation setting will benefit future work and challenge studies to look beyond shallow groundwater and investigate deeper groundwater. We have taken your suggestion and added the following text to the manuscript.

"In fact, the lack of subsurface characterization in high elevation groundwater systems often forces hydrologic modeling studies to focus solely on shallow groundwater systems and make the generalizing assumption that hydraulic conductivity decreases with depth and deep fractured aquifers support little flow (Manning and Caine, 2007; Welch and Allen, 2014; Markovich et al., Accepted)." (Included in Introduction, Lines 59-62)

"Further, we suggest that the deep subsurface flow paths observed in the JRB-CZO are likely a signature of snow dominated volcanic catchments transferable to other deeply fractured extrusive bedrock systems, which highlights the need to consider deeper groundwater processes in integrated hydrologic models." (Included in Conclusion, Lines 766-769)

2) We now refer to the site 1 deep fractured tuff aquifer as an aquifer system instead.

3) As you suggested, the calculation of dh/dz between the two most shallow wells yields a hydraulic head gradient of 1.1 m, just slightly greater than unit gradient. Without a fully screened well between the depths of 2D and 2C (total depth of wells ~7mbgs and ~31 mbgs, respectively), we do not have direct evidence of an unsaturated zone beneath the shallowest well and we did not measure saturation of cores collected during drilling because of the use of drilling fluids. However, we have included pictures of continuous cores here (from depths between wells 2D and 2C with increasing depth with increasing sample number) that display redoximorphic features commonly used to identify transitions in porous media saturation between the two wells, which suggests that the shallow well 2D is a likely a perched aquifer.

We now refer to the "perched aquifer" as shallow groundwater and note in the first mention of the shallow groundwater that it is likely perched. We have also added the following text to Section 2.2, Groundwater well completions to explain why we think it is likely perched.

[revised manuscript text omitted]